JCB Journal of Cell Biology

# Intracellular nanovesicles mediate α5β1 integrin trafficking during cell migration

Gabrielle Larocque[1], Daniel J. Moore[1], Méghane Sittewelle[1], Cansu Kuey[1], Joseph H.R. Hetmanski[2], Penelope J. La-Borde[1], Beverley J. Wilson[2], Nicholas I. Clarke[1], Patrick T. Caswell[2], and Stephen J. Royle[1]

**Membrane traffic is an important regulator of cell migration through the endocytosis and recycling of cell surface receptors such as integrin heterodimers. Intracellular nanovesicles (INVs) are transport vesicles that are involved in multiple membrane trafficking steps, including the recycling pathway. The only known marker for INVs is tumor protein D54 (TPD54/TPD52L2), a member of the TPD52-like protein family. Overexpression of TPD52-like family proteins in cancer has been linked to poor prognosis and an aggressive metastatic phenotype, which suggests cell migration may be altered under these conditions. Here, we show that TPD54 directly binds membrane and associates with INVs via a conserved positively charged motif in its C terminus. We describe how other TPD52-like proteins are also associated with INVs, and we document the Rab GTPase complement of all INVs. Depletion of TPD52-like proteins inhibits cell migration and invasion, while their overexpression boosts motility. We show that inhibition of migration is likely due to altered recycling of α5β1 integrins in INVs.**

## Introduction

Cell migration is important for many aspects of animal physiology, including the immune response, tissue integrity, and embryonic development. This process is tightly controlled, and any alterations can result in diseases such as inflammation or cancer (Hamidi and Ivaska, 2018; Friedl and Wolf, 2003). Membrane traffic is a key regulator of cell migration through the trafficking of cell surface receptors, including integrins, which bind the ECM. Integrins are endocytosed and recycled back to the cell surface in order to break and reestablish the cellular contacts with the ECM during migration. A prototypical example is α5β1 integrin, which binds to fibronectin. The molecular details of integrin trafficking pathways and their influence on cell motility are under active investigation (Wilson et al., 2018).

The identity and activation state of integrin heterodimers govern their trafficking and fate (Wilson et al., 2018). Rabs are master regulators of membrane traffic with >60 different Rabs in human cells, with each one mediating a trafficking step specifically (Wandinger-Ness and Zerial, 2014). Rabs are involved in every step of integrin traffic, as well as binding integrins directly or via one of the many Rab effector proteins (Pellinen et al., 2006; Caswell et al., 2008).

A new class of intracellular transport vesicle, termed intracellular nanovesicles (INVs), has recently been described (Larocque et al., 2020). INVs are involved in recycling and anterograde trafficking pathways. These vesicles are small (∼30 nm diameter) and highly dynamic and are associated collectively with ∼16 different Rab GTPases (Larocque et al., 2020). Among the Rabs present on INVs are Rab11a and Rab25, two Rabs well known for their involvement in integrin trafficking (Roberts et al., 2001; Caswell et al., 2007; Moreno-Layseca et al., 2019). INVs were discovered because of their association with TPD54, a member of the tumor protein D52-like protein family (TPD52, TPD53/TPD52L1, TPD54/TPD52L2, and TPD55/TPD52L3). How TPD54 associates with INVs and whether the other members of the TPD52-like protein family behave similarly are important open questions.

TPD52-like proteins were identified due to their over-expression in a number of cancer types (Byrne et al., 1995, 1996; Nourse et al., 1998; Cao et al., 2006). This up-regulation is often caused by gene duplication, which is tumorigenic (Balleine et al., 2000; Lewis et al., 2007). Tumorigenicity has been proposed to be due to alteration of either the cell cycle (Boutros and Byrne, 2005; Thomas et al., 2010; Lewis et al., 2007), signaling (Li et al., 2017), or DNA repair (Chen et al., 2013). Finally, TPD52-like proteins have been reported to have a role in cell migration and adhesion; however, the underlying mechanism is unknown (Ummanni et al., 2008; Mukudai et al., 2013). In breast cancer, TPD52 overexpression correlates with poor prognosis and a

[1]Centre for Mechanochemical Cell Biology, Warwick Medical School, Coventry, UK; [2]Wellcome Trust Centre for Cell-Matrix Research, Faculty of Biology, Medicine and Health, University of Manchester, Manchester Academic Health Science Centre, Manchester, UK.

Correspondence to Stephen J. Royle: s.j.royle@warwick.ac.uk; G. Larocque's present address is Cellular Signalling and Cytoskeletal Function Laboratory, The Francis Crick Institute, London, UK.



decrease in metastasis-free survival (Roslan et al., 2014; Shehata et al., 2008). This suggested to us that TPD52-like proteins, and the INVs they are associated with, may be involved in cell migration and invasion in cancer.

In this paper, we show how TPD52-like proteins associate with INVs and document the Rab complement of INVs decorated with TPD52, TPD53, or TPD54. We find that depletion of TPD52-like proteins inhibits cell migration and invasion, and we show that this is likely due to altered α5β1 integrin recycling via INVs.

## Results

### Molecular determinants required for the association of TPD54 with INVs

We previously found that TPD54 is tightly associated with INVs and that its association with these fast-moving subresolution vesicles could be measured by spatiotemporal variance of fluorescence microscopy images (Larocque et al., 2020). Here, we asked what are the molecular determinants for the association of TPD54 with INVs. Analysis of the primary sequences of TPD52-like proteins reveals two domains: (1) a coiled-coil domain between residues 38 and 82 (Fig. 1 A) and (2) a region between residues 126 and 180 with high similarity among human TPD52-like proteins. Within this region, residues 159–171 (Fig. 1 A) are particularly well conserved across different species. With these regions in mind, we designed mCherry-FKBP–tagged TPD54 constructs to pinpoint which region of the protein was required for its association with INVs (Fig. 1, B and C). The spatiotemporal variance of fluorescence for mCherry-FKBP-TPD54 full length was much higher than that of the control mCherry-FKBP, which is to be expected due to the association of TPD54 with mobile, subresolution vesicles such as INVs (Fig. 1 B and C; and Video 1). The spatiotemporal variance of TPD54 constructs revealed that the C-terminal portion of TPD54 (155–180) was needed for INV localization (Fig. 1 B). Constructs that lacked this region (1–37, 1–82, 38–82, and 1–155) had low variance, while those that contained it (1–206, 83–206, and 1–180) had higher variance. However, while this region was necessary for localization, it was not sufficient, since 155–180, 155–206, 126–180, and 126–206 did not have high spatiotemporal variance.

The region around residues 155–180 contains several positively charged residues (Fig. 1 A), suggesting that this region could bind direct to the INV membrane. Mutation of R95, K154, R159, K165, or K175/K177 to glutamic acid reduced the spatiotemporal variance to levels similar to control (Fig. 1, B and C). A similar mutation (K15E) in the N-terminal part of TPD54 had no significant effect. These experiments establish that positively charged residues around the conserved 155–180 region of TPD54 are important for its association with INVs.

### TPD54 directly binds liposomes in vitro

The association of TPD54 with INVs could either be direct or via another membrane-associated protein. We therefore investigated whether TPD54 was able to bind vesicles directly using a liposome-binding assay. From a Folch extract of brain lipids, we generated liposomes of four different sizes (30, 50, 100, and 200 nm) and tested for cosedimentation of GST, GST-TPD54, or GST-

TPD54 mutants with these liposomes (Fig. 2 A). We observed specific cosedimentation of GST-TPD54 proteins, but not GST alone, with liposomes of all sizes (Fig. 2 A). To quantify this direct membrane binding, we first measured the efficiency of liposome pelleting since smaller liposomes sediment less efficiently (Boucrot et al., 2012) and are less abundant (see Materials and methods; Fig. 2, B and C). Quantification of cosedimentation, accounting for liposome pelleting efficiency, revealed that GST-TPD54 bound 30 nm and 50 nm tighter than larger-diameter liposomes (Fig. 2, A and D). GST-TPD54(1–82) showed no binding and was indistinguishable from GST alone, indicating that the C-terminal portion of TPD54 was responsible for direct binding (Fig. 2 D). However, mutation of R169E, K154E, K175,177E, or K165E did not reduce membrane association in this assay. Indeed deletion of the C-terminal conserved region displayed similar binding to full-length TPD54 (1–155; Fig. 2 E). These results indicate that TPD54 can bind to membranes directly and that this is conferred by residues 83–155. However, the lack of effect of mutation of the C-terminal domain suggests that the liposomes in this assay are missing one or more factors that are present on INVs in cells.

### Positive residues in the conserved C-terminal region of TPD54 are required for INV association

To tackle the apparent discrepancy between spatiotemporal variance measurements in cells and in vitro liposome-binding experiments, we used an alternative method to test for association of TPD54 constructs with INVs. Previously, we found that rerouting an FKBP-tagged TPD54 construct to MitoTrap on mitochondria using rapamycin causes mitochondrial aggregation due to capture of INVs (Fig. 3, A and B; Larocque et al., 2020). We therefore tested the TPD54 constructs using this method and measured mitochondrial aggregation in confocal micrographs with automated image analysis (Fig. 3, C and D). Generally, the constructs that exhibited high spatiotemporal variance caused significant mitochondrial aggregation, whereas the mitochondria were unaffected by the rerouting of constructs that had showed low variance (Fig. 3, C and D). The only exceptions were R95E and K165E; rerouting of either mutant induced mitochondrial aggregation to some extent. The results from the two cellular assays underline the importance of positive residues K154, R159, K175, and K177 in the conserved C-terminal region of TPD54 for binding INVs. Moreover, residues 83–125 are also required for binding, since 83–206, but not 126–206, was associated with INVs (Figs. 1 C and 3 D). Altogether, our results indicate that there are two regions of TPD54 that are each necessary for association with INVs, but neither region is sufficient by itself. A region between 83–125 is required for membrane binding whereas positive charge in the conserved C-terminal domain is needed for specific association with INVs in cells.

### TPD52-like proteins can homo- and heteromerize via the coiled-coil domain, but multimerization is dispensable for INV binding

It is likely that TPD54 can form homomers or heteromers with other TPD52-like proteins via their coiled-coil domains (Byrne

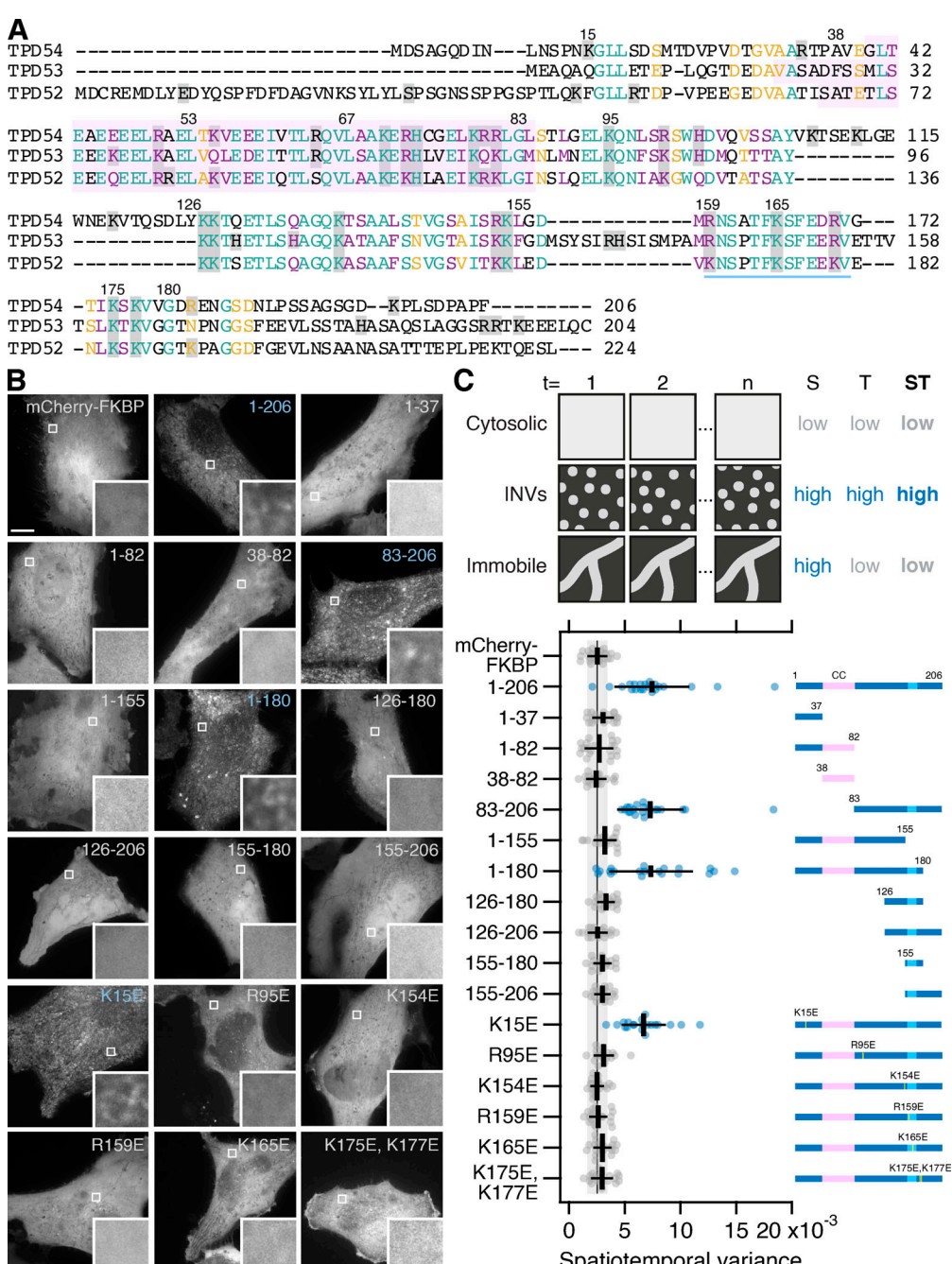

Figure 1. **Identification of the region required for the association of TPD54 with INVs. (A)** Alignment of human TPD54/TPD52L2 (UniProt accession no. O43399), TPD53/TPD52L1 (UniProt accession no. Q16890) and TDP52/TPD52 (UniProt accession no. P55327). Numbers indicate residue positions in TPD54. Pink highlighted area represents the coiled-coil domain (predicted with PCOILS, window size 28). Blue underlining indicates C-terminal conserved region. Gray shadowing shows positively charged residues. Lettering: teal, fully conserved residues; purple, strongly similar (>0.5 in the Gonnet PAM 250 matrix); and yellow, weakly similar (<0.5 in the Gonnet PAM 250 matrix). **(B)** Representative confocal micrographs of HeLa cells expressing mCherry-FKBP or mCherry-FKBP–tagged TPD54 FL (1–206) or indicated mutants. Blue labels indicate high variance measured in C. Inset: 4× zoom. Scale bar, 10 μm or 1 μm (inset). **(C)** Schematic diagram to show how spatiotemporal (ST) variance can be used to measure association with motile vesicular structures such as INVs. Spatial (S) and temporal (T) variance is shown for comparison. Scatter dot plot to show the spatiotemporal variance (mean variance per pixel over time) for the indicated constructs. Dots, individual cells; black bars, mean ± SD. The mean ± SD for mCherry-FKBP (control) is also shown as a black line and gray zone, down the plot. Dunnett's post hoc test was done using mCherry-FKBP as control; blue indicates P < 0.05. Right: Representation of the mCherry-FKBP–tagged TPD54 constructs analyzed. Pink region, coiled-coil domain (CC). Yellow line, position of point mutation. Light blue region, underlined area in A.

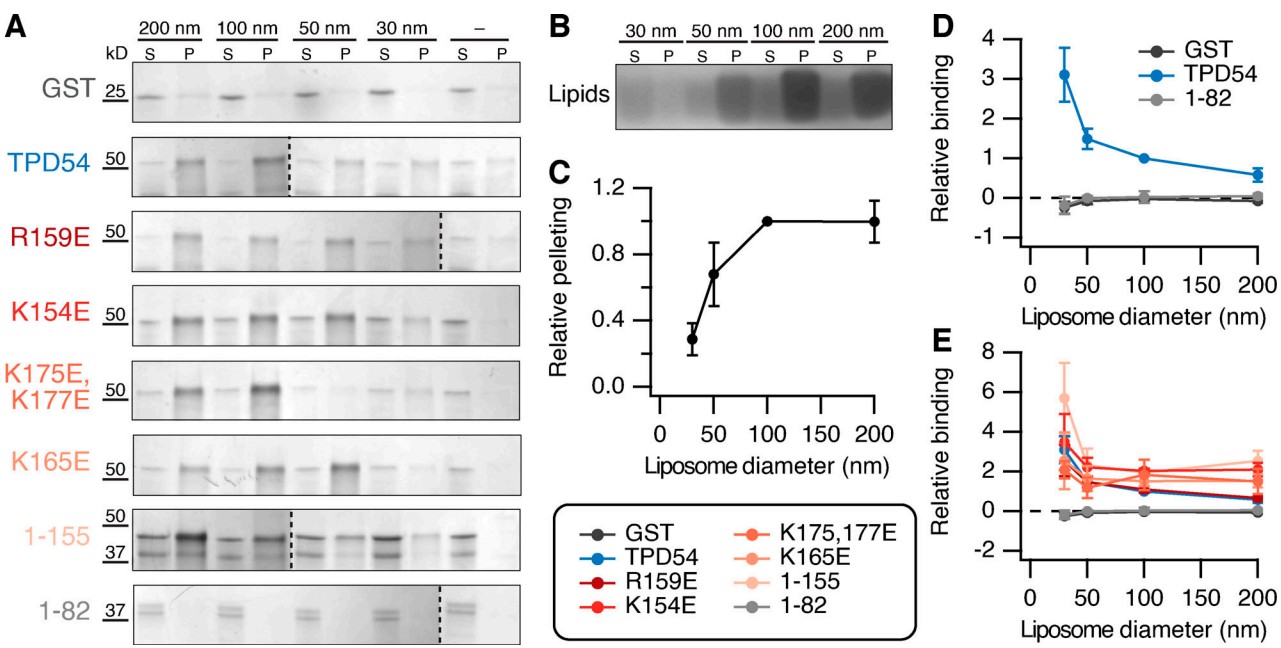

**Figure 2. TPD54 directly associates with liposomes in vitro. (A)** Cosedimentation of GST, GST-TPD54 WT, or GST-TPD54 mutants as indicated with differently sized liposomes (diameter indicated) or no liposomes (−), visualized on an InstantBlue-stained 4% to 12% gel. S, supernatant; P, pellet. **(B)** Pelleting of differently sized liposomes visualized on a Coomassie-stained Bis-Tris gel showing lipid stain. **(C)** Quantification of the pelleting efficiency of liposomes according to their size. Values are normalized to 100 nm liposomes. Points show mean ± SD; n = 3. **(D)** Quantification of protein cosedimentation with differently sized liposomes. The binding of GST (dark gray), GST-TPD54 (WT, blue), or GST-TPD54 (1–82, light gray) is shown relative to TPD54 binding to 100 nm. **(E)** Quantification as in D, but with all mutants overlaid. Points show mean ± SEM; n = 3, except n = 6 for GST, TPD54, and R159E.

et al., 1998; Sathasivam et al., 2001; Larocque et al., 2020). This raises the question of whether the association with INVs requires multimerization. To answer this point, we sought a TPD54 mutant that was incapable of multimerization. Mutation of two leucines to prolines is predicted to break the coiled-coil domain of TPD54 (L53P,L67P; Figs. 1 A and 4 A), and we tested whether these mutations interfered with homo- and heteromerization in cells. FLAG-tagged TPD52-like proteins were immunoprecipitated from cells coexpressing GFP-FKBP, GFP-FKBP-TPD54, or GFP-FKBP-TPD54(L53P,L67P). We found that TPD54 WT, but neither the control nor the mutant, could be coimmunoprecipitated with FLAG-tagged TPD52, TPD53, or TPD54 (Fig. 4 B). These results confirmed that the coiled-coil domain of TPD54 is responsible for homo- and heteromerization and that the L53P,L67P mutation blocks multimerization. Note that we assume that multimerization is direct between TPD52-like protein monomers, but our experiments do not rule out an intermediary protein. Next, we found that the localization of the L53P,L67P mutant was normal and its spatiotemporal variance was similar to WT TPD54, suggesting that it was associated with INVs (Fig. 4, C and D). This result is in agreement with the finding that a TPD54 construct lacking the coiled-coil region (83–206) is localized to INVs (Figs. 1 C and 3 D). Together these results indicate that monomeric TPD54 can associate with INVs and that multimerization is not required.

**TPD52 and TPD53 are associated with INVs**

Given the similarity of TPD52-like proteins and the conservation of the residues involved in INV association (Fig. 1 A), it is likely

that TPD52 and TPD53 are also found on INVs. Indeed, live-cell imaging indicates that they have a similar subcellular distribution and spatiotemporal variance (Video 2). The TPD54(L53P,L67P) mutant allowed us to ask whether other TPD52-like proteins are associated with INVs independently of TPD54. To do this, we used the mitochondrial INV-capture procedure using mCherry-FKBP-TPD54 or mCherry-FKBP-TPD54(L53P,L67P). We found that when GFP-TPD52 is coexpressed, it also becomes rerouted to the mitochondria (Fig. 5 A). This indicated that not only is TPD52 on INVs but also its association is likely direct and not via recruitment by TPD54. To confirm that TPD52 is an INV protein, we used correlative light EM (CLEM) to visualize the capture of INVs when GFP-FKBP-TPD52 was rerouted to mitochondria (Fig. 5 B).

To test whether TPD52 and TPD53 were bound to INVs independently of TPD54, we used the mitochondrial aggregation assay. Significant aggregation was seen for GFP-FKBP–tagged TPD52, TPD53, or TPD54 in control cells or those depleted of endogenous TPD54 by RNAi (Fig. 5, B and C). We noted that aggregation was less efficient for TPD53 compared with TPD52 or TPD54, suggesting that TPD53 is less efficiently targeted to INVs. These results confirm that TPD52-like proteins bind INVs and do so independently of TPD54.

**TPD52 and TPD53 are associated with different subsets of INVs**

INVs are involved in many trafficking pathways, since they collectively have a variety of Rab GTPases (Larocque et al., 2020). This was demonstrated by using mCherry-FKBP-TPD54 in a vesicle capture assay and asking which GFP-Rabs were

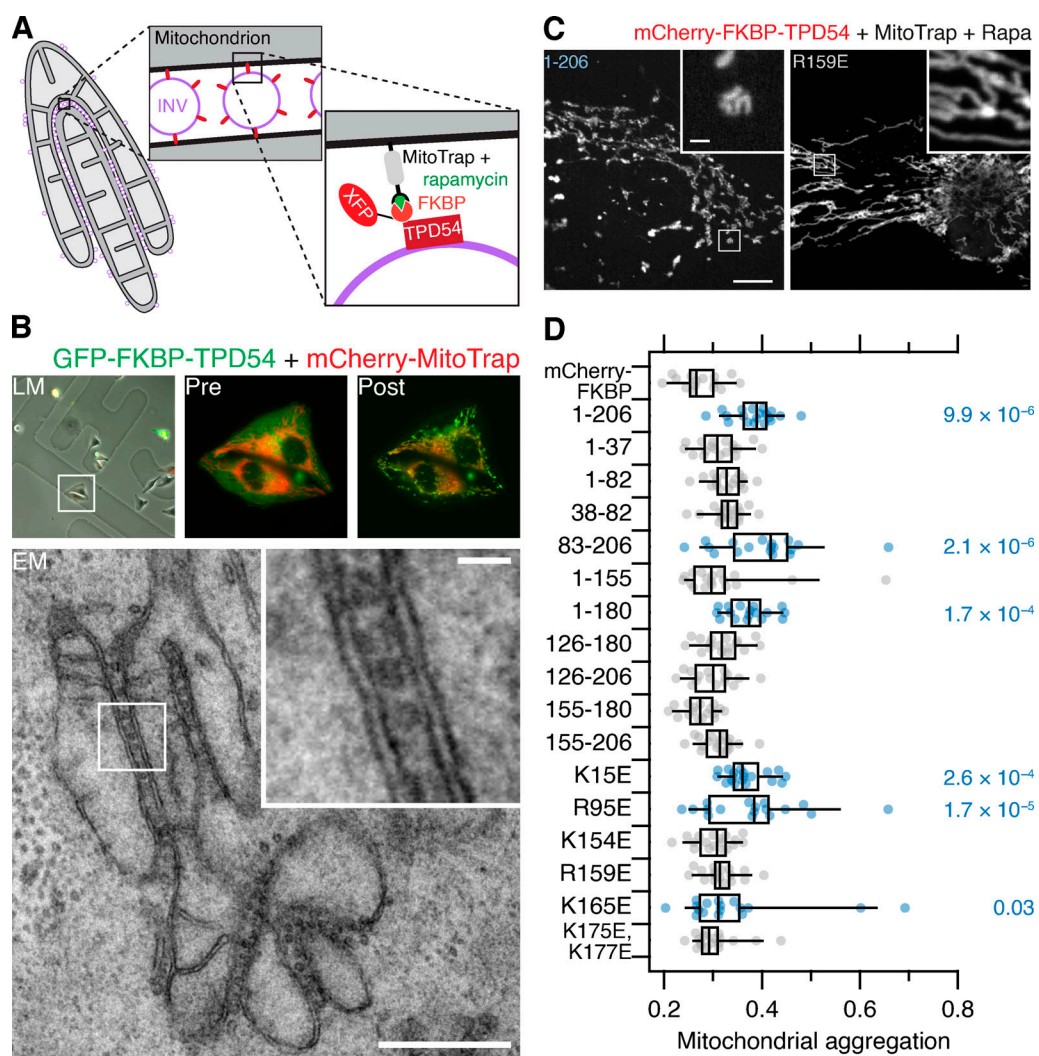

Figure 3. **INV-induced mitochondrial aggregation as an assay for TPD54 binding to INVs. (A)** Schematic diagram of vesicle capture at mitochondria and their subsequent aggregation. MitoTrap is an FRB domain targeted to mitochondria, XFP-FKBP-TPD54 (XFP is GFP or mCherry) is coexpressed, and, when rapamycin is added, the INVs associated with TPD54 become trapped at the mitochondria, eventually causing aggregation of mitochondria. **(B)** CLEM experiment to demonstrate INVs between aggregated mitochondria. Cells expressing GFP-FKBP-TPD54 and mCherry-MitoTrap were imaged before light microscopy (LM), before (Pre) and after (Post) rapamycin 200 nM addition for 3 min. An ultrathin section of the same cell is shown by transmission EM. Inset: 4× zoom. Scale bars, 500 nm or 50 nm (insets). **(C)** Representative confocal micrographs of HeLa cells expressing dark MitoTrap and either mCherry-FKBP-TPD54 WT (1–206) or R159E mutant, treated with 200 nM rapamycin. Inset: 5× zoom. Scale bars, 10 µm or 1 µm (inset). **(D)** Quantification of mitochondrial aggregation in cells expressing mCherry-FKBP–tagged TPD54 constructs and GFP-MitoTrap, treated with 200 nM rapamycin for 30 min. Dots show average mitochondrial shape (high values are more aggregated) per cell (see Materials and methods). Box plots show interquartile range (IQR), bar represents the median, and whiskers show 9th and 91st percentiles. Dunnett's post hoc test was done using mCherry-FKBP as control; blue indicates P < 0.05.

corerouted to mitochondria. Since TPD52 and TPD53 also are on INVs, we wanted to know if all INVs have TPD54 or if there are subsets of INVs with different TPD52-like proteins. To investigate this point, we performed mitochondrial vesicle capture and quantified the corerouting of GFP-Rabs to mitochondria using either mCherry-FKBP-TPD52 (Fig. 6, A and B) or mCherry-FKBP-TPD53 (Fig. 6, C and D). Of the 39 Rabs tested, 16 corerouted with TPD52 and 9 corerouted with TPD53. As with TPD54, Rab30 was the strongest hit with TPD52 (Fig. 6 B) and TPD53 (Fig. 6 D). This suggests that Rab30 is the Rab GTPase most likely to be associated with INVs and can be thought of as an "INV Rab." The other Rabs corerouted with TPD52 were Rab14, Rab26, Rab1a, Rab1b, Rab10, Rab17, Rab33a, Rab19b,

Rab4a, Rab3a, Rab25, Rab21, Rab12, and Rab43. Of these, only Rab21 had not been identified in the TPD54 screen (Larocque et al., 2020). The Rabs corerouted with TPD53 were Rab30, Rab1b, Rab26, Rab1a, Rab33b, Rab43, Rab19b, Rab14, Rab12, and Rab10; all of which also corerouted with both TPD54 and TPD52. To classify INVs in a more stringent manner, we used hierarchical clustering of the mean mitochondrial intensity change of GFP-Rabs in the TPD52 and TPD53 vesicle capture screens (Fig. 6, A and C), together with the previously published TPD54 screen (Fig. 7 A). This resulted in the classification shown in Fig. 7 B.

The GFP-Rabs that were top hits in the vesicle capture screens all had subcellular distributions at steady state that were

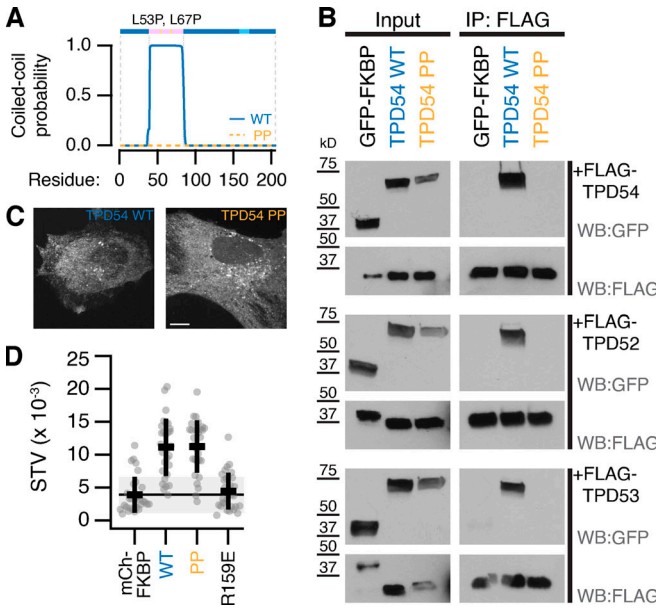

**Figure 4. TPD54 homo- and heteromerizes using its coiled-coil domain.**
**(A)** Schematic representation of TPD54 and a graph showing the coiled-coil probability for WT and L53P,L67P (PP) mutant. Pink, coiled-coil domain; light blue, conserved region. For coiled-coil domain prediction, the amino acid sequences of TPD54 isoform 1 WT or TPD54(L53P,L67P) were analyzed by PCOILS with window size of 28 (Gruber et al., 2006). **(B)** Western blot (WB) showing immunoprecipitation (IP) of FLAG-TPDs and coimmunoprecipitation of GFP-FKBP, GFP-FKBP-TPD54 (WT) or GFP-FKBP-TPD54(L53P,L67P; PP). **(C)** Representative confocal image of mCherry-FKBP-TPD54 (TPD54 WT) or mCherry-FKBP-TPD54(L53P,L67P; TPD54 PP) expressed in HeLa cells. Scale bar, 10 μm. **(D)** Scatter dot plot to show the spatiotemporal variance (STV) of the indicated constructs. Dots, individual cells; black bars, mean ± SD. The mean ± SD for mCherry-FKBP (control) is also shown as a black line and gray zone. Values for R159E are shown as a negative control. WT and PP, P = 1.9 × $10^{-6}$; Dunnett's post hoc test using mCherry-FKBP as control.

similar to that of TPD52-like proteins (Video 3). This suggested that if we quantify the spatiotemporal variance of individual Rabs, we could determine which Rabs are associated with INVs, to verify the results from the vesicle capture screens. Broadly, the Rabs that corerouted with one or more TPD52-like protein had high spatiotemporal variance and those that did not had lower variance (Fig. 7 C). This analysis verified that these Rabs were present on INVs that were positive for TPD52-like proteins. In theory, this approach could also be used to identify INVs that lacked TPD52-like proteins. Indeed, high variance was seen for Rab18, Rab9a, and Rab15, although none of these Rabs were corerouted with any of the three TPD52-like proteins tested (Fig. 7 C). Close inspection of these movies, however, suggest that the high variance of these Rabs is likely a false positive, since they are present on larger structures and not subresolution vesicles.

In summary, this analysis suggests that there are four INV populations. The first population has all three TPD52-like proteins and either Rab30, Rab14, Rab1a, Rab1b, or Rab26. The second population is identified by a predominance of TPD52 and Rab10 or Rab17; the third by the predominance of TPD54 and Rab4a, Rab25, or Rab3a; and the fourth by TPD53 and Rab33b or Rab19b. There are also intermediates (TPD52 and/or TPD54 with Rab11a and TPD53 and/or TPD54 with Rab12 or Rab43; Fig. 7 B).

Taken together, the data highlight the existence of different populations of INVs, marked by various Rabs, but all characterized by the presence of at least one member of the TPD52-like protein family.

### Amplification of TPD52-like proteins in cancers and potential changes in cell migration

Previous work showed that TPD52-like proteins are overexpressed in several cancer types (Nourse et al., 1998; Byrne et al., 1996), which may be associated with a more metastatic phenotype (Roslan et al., 2014; Shehata et al., 2008; Ummanni et al., 2008; Mukudai et al., 2013). To provide a full picture of TPD52-like protein overexpression in cancer, we analyzed The Cancer Genome Atlas PanCancer Atlas (Fig. S1). Amplification of TPD52 and TPD54 was seen in a range of cancers including breast invasive carcinoma, ovarian and uterine cancers, and cancers of the colon and liver. Amplification of TPD53 and TPD55 was less common (Fig. S1 A). Rabs that were associated with INVs and those that were not were also analyzed. Of these, the INV-associated Rab25 had a similar amplification profile to TPD52-like proteins (Fig. S1, B and C). Analysis of the ovarian serous carcinoma dataset showed that of 398 patients, amplification of TPD54 was seen in 29 (7%), amplification of Rab25 in 20 (5%), and a significant cooccurrence in 7 patients (log2 odds ratio >3, P < 0.001, and q < 0.001). These results prompted us to investigate a potential link between TPD52-like proteins and cell migration and invasion.

### TPD52-like proteins are important for 2D and 3D cell migration

We imaged control RPE1 cells and those depleted of TPD54, as they migrated on two different 2D substrates, fibronectin (Fig. 8 and Video 4) or laminin (Fig. S2). Tracking cells over 12 h allowed us to generate a complete assessment of their migratory behavior (Fig. S3). We found that cells depleted of TPD54 by RNAi had a strong reduction in migration speed on both substrates, indicating that this defect is not restricted to a single integrin heterodimer (Huttenlocher and Horwitz, 2011; Figs. 8 A and S2). To check whether the TPD54 RNAi phenotype was the result of an off-target effect, we assessed migration speed on fibronectin of cells treated with a further two TPD54-targeting siRNAs (Fig. 8 B). A similar reduction was seen with all three siRNAs compared with control. We also tested whether the TPD54 RNAi phenotype could be rescued. We compared cells transfected with siCtrl expressing GFP with those transfected with siTPD54 that reexpressed GFP or RNAi-resistant GFP-TPD54 WT (Fig. 8 C). The migration defect was indeed rescued by reexpression of GFP-TPD54, but not GFP alone. More importantly, we wanted to test if the ability to bind to INVs was necessary for the role of TPD54 in migration. TPD54-depleted cells reexpressing a construct that lacked the INV-binding region (GFP-TPD54 1–155) failed to rescue the migration defect (Fig. 8 C). This shows that it is the ability to bind INVs that allows TPD54 to rescue the migration phenotype and implicates INVs in cell migration.

Since depletion of TPD54 reduced migration and overexpression is linked to disease, we next tested the effect of TPD54 overexpression. Overexpression of TPD54 caused a

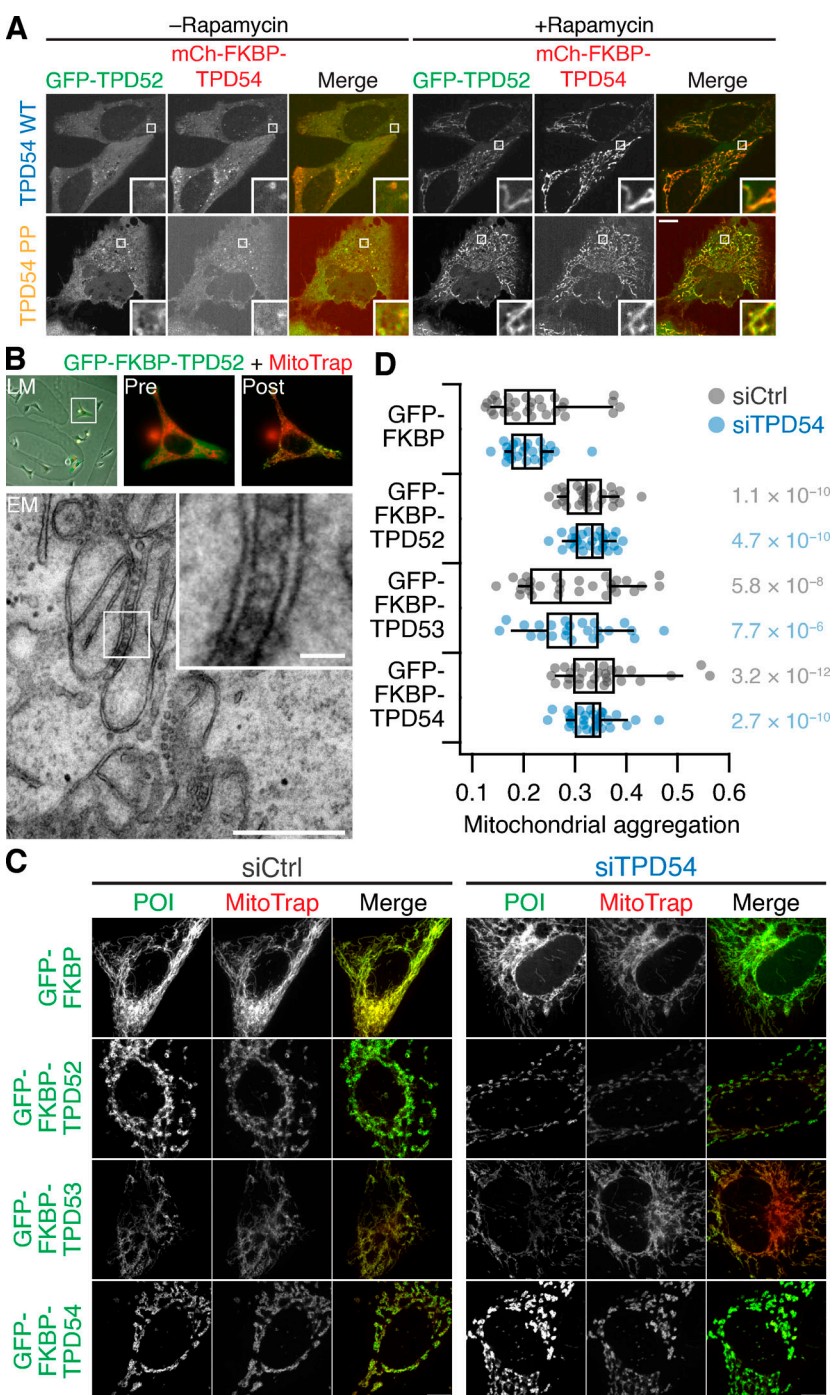

**A** | −Rapamycin | +Rapamycin
mCh-FKBP-TPD54 | mCh-FKBP-TPD54
TPD54 WT | GFP-TPD52 | mCh-FKBP-TPD54 | Merge | GFP-TPD52 | mCh-FKBP-TPD54 | Merge
TPD54 PP

**B** GFP-FKBP-TPD52 + MitoTrap
LM | Pre | Post
EM

**C** siCtrl | siTPD54
POI | MitoTrap | Merge | POI | MitoTrap | Merge
GFP-FKBP
GFP-FKBP-TPD52
GFP-FKBP-TPD53
GFP-FKBP-TPD54

**D**
GFP-FKBP
GFP-FKBP-TPD52 — $1.1 \times 10^{-10}$ / $4.7 \times 10^{-10}$
GFP-FKBP-TPD53 — $5.8 \times 10^{-8}$ / $7.7 \times 10^{-6}$
GFP-FKBP-TPD54 — $3.2 \times 10^{-12}$ / $2.7 \times 10^{-10}$
siCtrl / siTPD54
Mitochondrial aggregation (0.1 0.2 0.3 0.4 0.5 0.6)

Figure 5. **Vesicle capture and mitochondrial aggregation using rerouting of TPD52-like proteins. (A)** Representative confocal micrographs showing the co-rerouting of GFP-TPD52 after rerouting of mCherry-FKBP-TPD54 (top) or mCherry-FKBP-TPD54(L53P,L67P; PP; bottom) to dark MitoTrap by addition of 200 nM rapamycin. Inset: 5× zoom. Scale bar, 10 µm. **(B)** CLEM experiment to test for vesicle capture using TPD52. Cells expressing GFP-FKBP-TPD52 and mCherry-MitoTrap with rapamycin 200 nM for 3 min. Insets: 4× zoom. Scale bars, 500 nm or 50 nm (inset). **(C)** Representative confocal micrographs showing rerouting and mitochondrial aggregation. GFP-FKBP (control) or GFP-FKBP-TPD52-like (POI, green) proteins were coexpressed with mCherry-MitoTrap (red) in cells treated with control siRNA (siControl) or TPD54 siRNA as indicated and treated with 200 nM rapamycin for 30 min. Scale bar, 10 µm. **(D)** Quantification of mitochondrial aggregation. Dots show average mitochondrial shape per cell. Box plots show IQR, and bar represents the median and whiskers show 9th and 91st percentiles. Right: P values from Dunnett's post hoc tests using respective GFP-FKBP as control.

significant increase in migration speed of RPE-1 cells on fibronectin compared with expression of GFP alone (Fig. 8 D). No change in speed was observed when the R159E mutant was overexpressed again, suggesting that altered migration depended on INV localization of TPD54.

As we had established that TPD52 and TPD53 are also associated with INVs, we hypothesized that their depletion would also affect migration of cells on fibronectin (Fig. 8, E–G; and Fig. S4). When compared with controls, TPD52-depleted RPE1 cells also showed a significant reduction in migration speed (Fig. 8 E). Similarly, TPD53-depleted cells show a decrease in speed, but there was more variation between experiments, which suggests

a minor role for TPD53 in cell migration in comparison with TPD52 and TPD54, which likely reflects their relative abundance (Fig. 8 F). In addition, overexpression of either TPD52 or TPD53 increased migration speed compared with expression of GFP alone (Fig. 8 G). These experiments argue for functional redundancy among the TPD52-like protein family. When compared with knockdown of TPD54, the dual knockdown of TPD52 and TPD54 or TPD53 and TPD54 did not show an additive effect on migration speed (siGL2, 0.40; si54, 0.32; si52/54, 0.35; and si53/54, 0.36 µm min⁻¹; number of experiments [$n_{exp}$] = 3), suggesting that TPD54 dominates the role of these proteins in cell migration. In summary, TPD52-like proteins have a role in

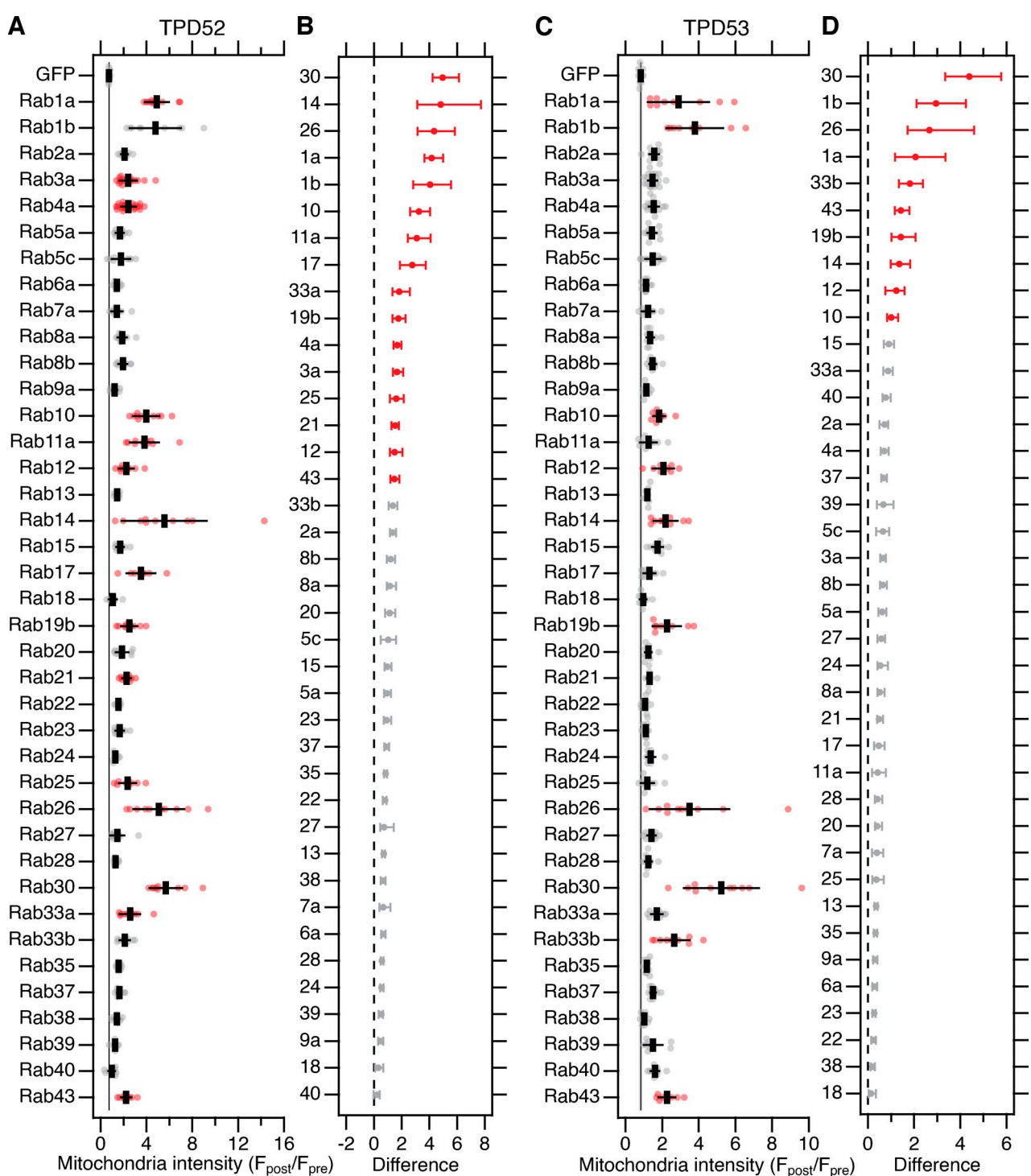

Figure 6. **Screening Rab GTPases that are associated with TPD52 and TPD53 INVs. (A and C)** Quantification of the change in mitochondrial fluorescence intensity of GFP or GFP-Rabs 2 min after rerouting of mCherry-FKBP-TPD52 (A) or mCherry-FKBP-TPD53 (C) to dark MitoTrap with 200 nM rapamycin. $F_{post}$/$F_{pre}$ is the average fluorescence of indicated GFP-Rab at mitochondria after rerouting ("post") divided by the fluorescence in the same region before ("pre"). Dots represent values for individual cells across three independent trials. Black bars represent mean ± SD. The mean ± SD for GFP (control) is also shown as a black line and gray zone, down the plot. Red indicates P < 0.05; Dunnett's post hoc test, GFP as control. **(B and D)** Effect size and bootstrap 95% confidence interval of the data in A and C, respectively.

2D cell migration, and changes in their expression level can cause changes in migration speed; moreover, INVs are implicated in this behavior, since TPD54 mutants that cannot bind INVs are unable to modulate migration speed.

There are important differences between cells migrating on a flat substrate and those invading a 3D structure, and the latter is a more accurate model for cell movement in a cancer context. We therefore wanted to determine if TPD52-like proteins were

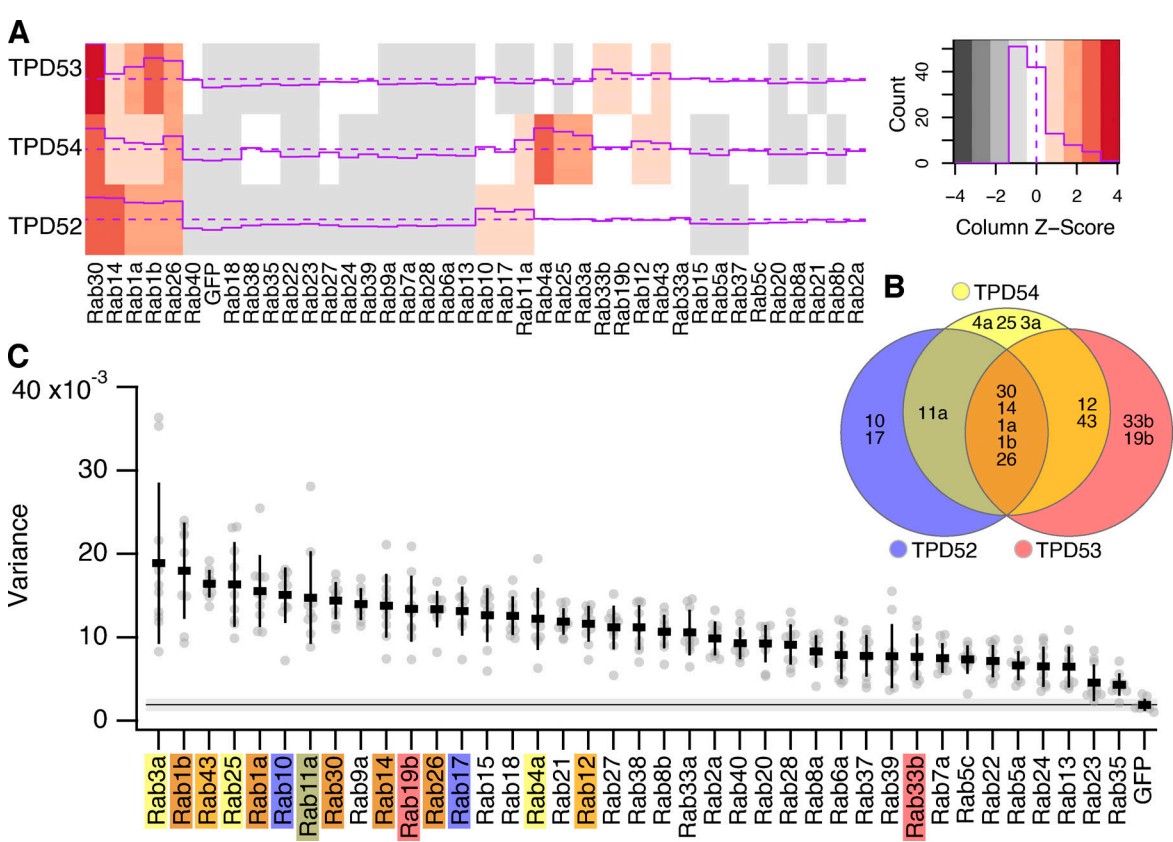

Figure 7. **Spatiotemporal variance of GFP-Rab proteins. (A)** Heatmap showing the Rab screen data for TPD52, TPD53 (from Fig. 6, A and C), and TPD54 (Larocque et al., 2020). Z-score of the values is color-coded and depicted as a purple line. Distribution of Z-values is shown in the color key. **(B)** Euler plot to show the Rabs identified in the screens and how they are linked to TPD52-like proteins. **(C)** Scatter dot plot of variance of fluorescence for the indicated GFP-Rab proteins expressed in HeLa cells. Rabs identified as present on INVs by hierarchical clustering of vesicle capture screens are indicated by the colors indicated in B. Dots represent individual cells, and bars indicates the mean ± SD. The mean ± SD for GFP (control) is also shown as a black line and gray zone, across the plot.

also important for cellular invasion. A2780 ovarian carcinoma cells that stably express Rab25 are an accurate model of an aggressive tumor (Cheng et al., 2004), and Rab25 increases invasion in a 3D microenvironment. First, we measured migration of these cells in cell-derived matrix (CDM), which is an elastic 3D matrix composed of fibrillar collagen and fibronectin. Depletion of TPD52 or TPD54, but not TPD53, caused a reduction in migration speed in CDM compared with control (Fig. 8 H and Video 5). Second, we tested the ability of these cells to invade a fibronectin-enriched collagen matrix for 48 h (Fig. 8 I). We quantified the invasion by measuring the total area that cells occupy deeper than 45 µm into the matrix. Again, cells depleted of TPD52 or TPD54, but not TPD53, lost their ability to invade a dense 3D matrix (Fig. 8 J). Taken together, the data show that TPD52-like proteins are important for cell migration both in noncancerous RPE1 cells migrating on a 2D substrate and in cancer cells invading a 3D structure.

**TPD54-depleted cells make larger contacts with their substrate**

In addition to changes in cell motility, we observed that TPD54-depleted cells seemed morphologically different from control cells (Fig. 9 A). To quantify this difference, we analyzed a series of cell shape parameters using a semiautomated workflow (see Materials and methods; Fig. S5). TPD54-depleted cells had much larger footprints, with an average area that was almost twice that of control cells (Fig. 9 B). Close inspection of movies of cells migrating on fibronectin revealed that the larger area and reduced speed of migration were linked. TPD54-depleted cells behaved as if they were "stuck" to the substrate; instead of having one clear lamellipodium, they made several smaller ones (Video 4). We hypothesized that these results may be due to a defect in integrin trafficking.

**INVs are involved in recycling α5β1 integrin**

To examine the possibility that integrins are trafficked in INVs, we sought to identify them using two proteomic approaches. First, we used mass spectrometry analysis of material immunoprecipitated from RPE-1 cells expressing GFP or GFP-TPD54 lysed under nonstringent conditions. TPD52 and TPD53 were detected alongside TPD54, as well as 19 Rab GTPases that were all, with the exception Rab7A, enriched in the TPD54 sample (Fig. 10 A). Several integrins were detected by this method, including α5β1, which was enriched compared with GFP. Second, identified proteins that were in close proximity to TPD54 on INVs. To do this, we compared proximity biotinylation

**Figure 8. Role for TPD52-like proteins in cell migration and invasion. (A–G)** RPE1 cell migration on a flat fibronectin substrate. **(A, E, and F)** Superplots showing migration speed of control vs TPD54- (A), TPD52- (E), or TPD53-depleted (G) cells. Dots represent individual cells, color-coded for experiments. Markers indicate mean speed for individual experiments. P value, Student's *t* test; $n_{exp}$ = 4 (TPD54), 3 (TPD52 and TPD53); whereas $n_{cell}$ = 469 (TPD54), 564,573 (TPD52), and 588,597 (TPD53). **(B)** Boxplot to show the migration speed of cells that were treated with siCtrl or each one of three TPD54-targeting siRNAs. Boxes show IQR, bar represents the median, and whiskers show 9th and 91st percentiles. P values from Dunnett's *post-hoc* test using siCtrl as control; $n_{cell}$ = 88–99, $n_{exp}$ = 1. **(C)** Violin plot showing the average speed of siCtrl- or siTPD54-treated cells expressing GFP, GFP-TPD54 WT, or GFP-TPD54 1–155 as indicated. Dots, individual cells; markers, mean speed. P values from Dunnett's post hoc test using siCtrl + GFP as control. $n_{cell}$ = 65–103, $n_{exp}$ = 1. **(D and G)** Superplots showing migration speed of cells expressing GFP, GFP + TPD54, or GFP + R159E (D) or GFP, GFP + TPD52, or GFP + TPD53 (E). P values in D from Dunnett's post hoc test using GFP as control. D, $n_{cell}$ = 247–259, $n_{exp}$ = 3. G, $n_{cell}$ = 129–131, $n_{exp}$ = 2. **(H–J)** Invasion of A2780 cells stably expressing Rab25 in a 3D context. **(H)** Boxplot of migration speed of cells in CDM that were treated with siRNAs as indicated. P values, Dunnett's post hoc test using siCtrl as control. $n_{exp}$ = 3. **(I)** Representative confocal images of A2780 cells stably expressing Rab25 treated with the indicated siRNAs migrating through fibronectin-supplemented collagen type-I matrix for 72 h. Scale bar, 250 μm. **(J)** Quantification of A2780 cell invasion in confocal sections ≥45 μm, normalized to siCtrl. Dots, individual wells. Box plots show IQR, bar represents the median, and whiskers show 9th and 91st percentiles. P value, Kruskal–Wallis test. $n_{exp}$ = 3.

profiles using either TPD54-HA-BioID2 WT or R159E mutant that does not localize to INVs. In this analysis, ITGA5 was only found in proximity with TPD54 and not the R159E mutant, while ITGB1 was enriched with TPD54 WT (Fig. 10 B). The presence of α5β1 integrins in INVs is consistent with the migration phenotypes observed on fibronectin or fibronectin-containing matrices.

Given the previously described role of INVs in recycling receptors, we therefore asked if recycling of α5β1 integrins is

affected by TPD54 depletion (Larocque et al., 2020). Using an ELISA-based recycling assay, we found that TPD54-depleted cells show a marked reduction in recycling of endocytosed α5 integrin heterodimers compared with control cells (Fig. 10 C). The initial surface label and uptake was similar in control and TPD54-depleted cells (98% and 84% of control, respectively). Aberrant traffic of internalized α5 integrin uptake in TPD54-depleted RPE-1 cells could be seen by labeling surface integrins

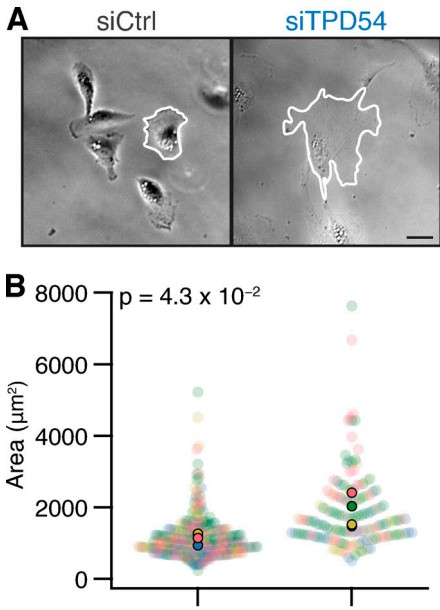

**Figure 9. TPD54-depleted cells make larger contacts with their substrate. (A)** Example micrographs of control and TPD54-depleted RPE1 cells migrating on fibronectin. The perimeter of the cell is outlined (white). Scale bar, 10 μm. **(B)** Superplot showing the area of control versus TPD54-depleted RPE1 cells migrating on fibronectin. Dots represent individual cells, colors represent different experiments, and the population mean is outlined in black. P value from Student's test. $n_{cell}$ = 288,151; $n_{exp}$ = 4. More statistics of cell shape are shown in Fig. S5.

and monitoring uptake and subsequent recycling (Fig. 10 D). TPD54-depleted cells showed accumulation of labeled integrins in intracellular compartments after 90 min consistent with a defect in recycling. Again, the initial surface (–30 min) and uptake (0 min) pools were similar (Fig. 10 D). These data confirm that TPD54 and the INVs are involved in trafficking of α5β1 integrin and provide a mechanistic explanation for the cell migration and invasion phenotypes observed in cells depleted of TPD52-like proteins.

## Discussion

In this study, we demonstrate the involvement of INVs in α5β1 integrin trafficking and cell migration. We describe how the TPD52-like protein TPD54 is localized to INVs and show that other members of this family, TPD52 and TPD53, are also INV proteins. This allowed us to document the Rab GTPase complement of INVs, which included Rabs that are involved in integrin traffic and cell migration. Depletion of TPD52-like protein family members caused decreased cell migration and invasion, a phenotype that could be linked to decreased integrin recycling.

The binding of TPD52-like proteins to INVs can be explained by two molecular properties. First, residues 83–125 of TPD54 are required for direct membrane binding. Second, positively charged residues in a C-terminal region that is conserved across metazoans govern the association with INVs. Both of these properties are necessary for TPD54 to bind INVs, but they are interdependent, such that neither property by itself is sufficient

for binding. Precisely what factor TPD52-like proteins recognize on INVs is an interesting question. Our in vitro liposome-binding data allowed us to reconstitute the general interaction between TPD54 and membranes but failed to recapitulate INV binding, since mutations that prevent association with INVs in cells had no effect in vitro. It is possible that the lipid composition of Folch extract did not match that of INVs. The small size of these vesicles may be due to an unusual lipid content that assists their formation (Kozlov et al., 2014). We favor this explanation rather than a protein factor that would itself require an explanation of how it associates with INVs. Future work on the purification of INVs from cells will allow their lipid and protein composition to be determined.

We exploited the direct binding of TPD52-like proteins to INVs in a mitochondrial vesicle capture assay in order to detail the collective Rab GTPase complement of INVs. This analysis showed that the TPD52-like proteins associate with similar INVs as delineated by their Rab GTPases. The Rabs associated with INVs cover anterograde, Golgi, and recycling trafficking, with the addition of Rab21, an endocytic Rab, which was captured using TPD52 only (Simpson et al., 2004). This Rab is notable here, since it is involved in the internalization of integrins (Pellinen et al., 2006). Rab30, a Golgi-resident Rab that is not fully characterized, was our top hit in vesicle capture using TPD52, TPD53 or TPD54; by this definition it could be considered as the most likely INV Rab. Rab30 was not among the Rabs identified in our proteomic analyses of INVs in RPE-1 cells, but it could be that expression of Rab30 is below detection in this cell line. Among the Rab30 effectors identified in a screen in *Drosophila melanogaster* were the dynein adaptor Bicaudal D and the tethering factors Golgi-associated retrograde protein (GARP) and exocyst (Gillingham et al., 2014), suggesting that Rab30 is involved in membrane fusion between the endosomes and the Golgi and between the Golgi and the plasma membrane. Rab30 has been shown to bind to PI4KB in an autophagy context (Oda et al., 2016; Nakajima et al., 2019). Whether Rab30 and PI4KB are needed for the formation or transport of INVs or whether GARP and exocyst are tethering factors present on INVs remains to be determined. Although limited to the 39 Rabs tested, we found no evidence for Rab-positive INVs that had no TPD52-like protein. This suggests that TPD52-like proteins are core components of INVs and possibly should be considered the molecules that define this class of transport vesicle.

We found that depletion of TPD54, TPD52, and, to a lesser extent, TPD53 decreased cell migration and invasion. Conversely, their overexpression increased 2D migration speed. These findings echo the literature showing that TPD52-like proteins are overexpressed in various cancers and that overexpression is potentially correlated with a more invasive, migratory phenotype (Roslan et al., 2014; Shehata et al., 2008). INVs are implicated in cell migration not only because depletion of a core INV protein impaired migration speed but also because normal motility could only be rescued by WT TPD54 and not by a truncated form that cannot bind to INVs. Similarly, overexpression of WT TPD54, which can bind INVs, caused an upregulation of migration speed, but a R159E mutant that does not bind INVs caused no up-regulation. Of the Rabs that we found

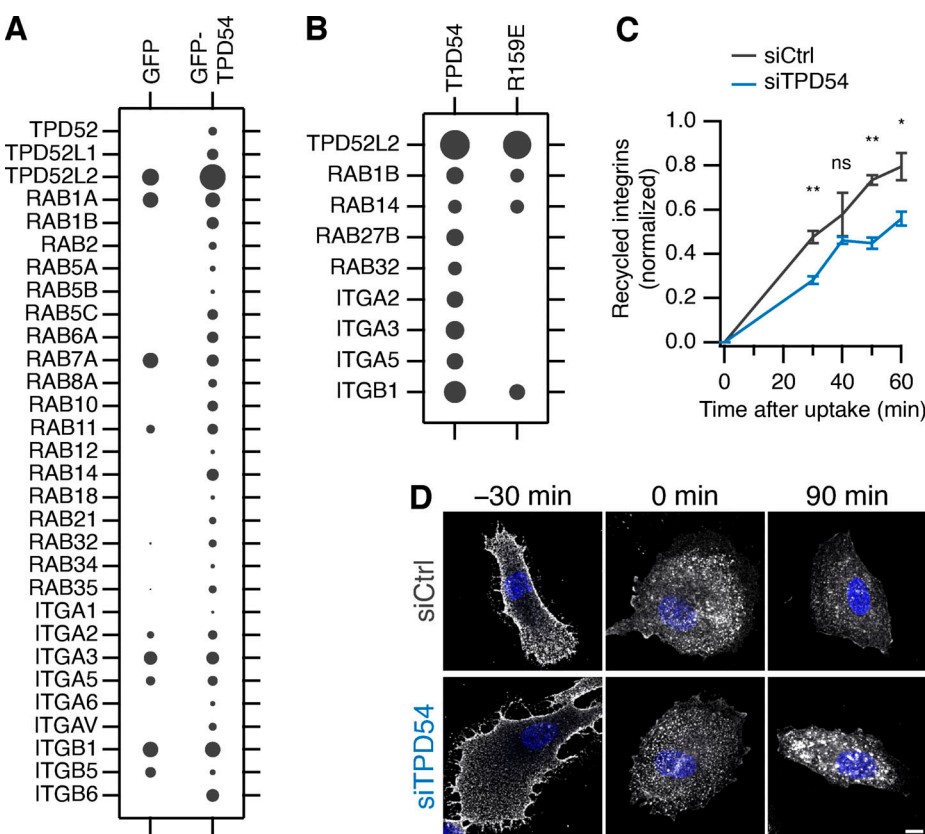

Figure 10. **INVs are involved in recycling α5β1 integrin. (A)** Bubble plot of total intensities from immunoprecipitates prepared from RPE-1 cells expressing GFP or GFP-TPD54 ($n_{exp}$ = 3). **(B)** Bubble plot of total spectral counts from BioID2 experiments ($n_{replicates}$ = 6, $n_{exp}$ = 2). In A and B, all TPD52-like proteins, Rabs and integrins detected in the dataset are shown. Size of bubbles is normalized to the most abundant protein detected per experiment. **(C)** ELISA-based quantification of integrin α5 recycling over time in siCtrl (gray line) or siTPD54-treated (blue line) RPE1 cells. **, P < 0.001; *, P < 0.05. Bars show SD. $n_{exp}$ = 3. **(D)** Representative confocal micrographs of integrin α5 recycling. RPE-1 cells (siCtrl or siTPD54 treated) were surface labeled using anti-integrin α5 (VC5) and allowed to recycle for the indicated times. The experiment was repeated three times, and similar results were also obtained using an alternative α5 antibody (SNAKA51). Scale bar, 10 µm.

associated with INVs, Rab4, Rab11, and Rab25 are all involved in integrin trafficking and cell migration (Caswell et al., 2007; Powelka et al., 2004; Roberts et al., 2001). After internalization, the integrin heterodimers are recycled back to the plasma membrane via a short, Rab4-dependent pathway or a long, Rab11-dependent pathway (Roberts et al., 2001); and in cancer cells, Rab25 sorts ligand-free integrins for recycling at the leading edge and ligand-bound integrins to lysosomes, where they reach the plasma membrane and cell rear in a CLIC3-dependent manner (Dozynkiewicz et al., 2012). Previously, we found that recycling of internalized transferrin was affected by depletion of TPD54, implicating INVs in receptor recycling (Larocque et al., 2020). Here, we showed that recycling of α5 integrin is impaired when TPD54 was depleted, suggesting that INVs also mediate recycling of internalized integrins. This raises an interesting question: if the size of INVs is invariant (~30 nm diameter), then how many integrin heterodimers can be trafficked in an INV? When the integrins are recycled to the plasma membrane, they are in their bent, inactive conformation. For integrin αIIbβ3, this conformation extends 11 nm (Ye et al., 2008), which indicates that traffic in INVs is possible. Despite their small size, the maximum theoretical capacity of an INV is

surprisingly high (Martins Ratamero and Royle, 2019 *Preprint*), although for bulky cargoes such as integrins, the number traveling in each INV is likely to be low.

## Materials and methods
### Molecular biology
Several plasmids were available from previous work, including mCherry-FKBP-TPD54, GFP-FKBP-TPD54, mCherry-MitoTrap, and dark MitoTrap (pMito-dCherry-FRB; Larocque et al., 2020). TPD52-like protein constructs used in the paper represent the canonical UniProt sequence (TPD52, P55327-1; TPD53, Q16890-1; and TPD54, O43399-1). To make mCherry-FKBP-TPD52, human TPD52 (synthesized by GeneArt) was inserted in place of TPD54 in mCherry-FKBP-TPD54 using BglII and MfeI. FLAG-TPD52 was made by amplifying human TPD52 from the synthesized gene and inserting into pFLAG-C1 via BglII and MfeI. To make mCherry-FKBP-TPD53, TPD53 (GeneArt synthesis) was inserted into pmCherry-FKBP-C1 via HindIII and BamHI. FLAG-TPD53 was made by amplifying TPD53 from the synthesized gene and inserting into pFLAG-C1 via BglII and BamHI. The plasmid to coexpress untagged TPD54 and GFP (pIRES-EGFP-TPD54) was

made by amplifying TPD54 by PCR from human tumor protein D54 (IMAGE clone 3446037) and inserting into pIRES-EGFP-puro (Addgene; 45567) via NheI and XhoI. Similar pIRES-EGFP-TPD52 and pIRES-EGFP-TPD53 were made by amplification from respective mCherry-FKBP–tagged constructs and insertion at NheI and XhoI. The mCherry-FKBP-TPD54 deletions (1–37, 1–82, 38–82, 83–206, 1–155, 1–180, 126–180, 126–206, 155–180, and 155–206) were made by PCR from mCherry-FKBP-TPD54 and were each inserted into pmCherry-FKBP-C1 via BglII and MfeI. The mCherry-FKBP-TPD54 mutants (K15E, R95E, K154E, R159E, K165E, K175E, and K177E) and GFP-FKBP-TPD54 mutant (L53P, L67P) were created by site-directed mutagenesis. GST-tagged TPD54 constructs were made by amplification from the mCherry-FKBP–tagged version and insertion in pGEX-6P-1. Plasmids to express GFP-tagged Rabs were a gift from Francis Barr (University of Oxford, Oxford, UK), except for GFP-Rab1a and GFP-Rab5c, which were described previously (Larocque et al., 2020). Plasmids to express HA-BioID2-TPD54 WT or R159E were made by inserting HA-BioID2 in place of GFP in the GFP-TPD54 at AgeI and BglII and then subsequently lengthening the linker between BioID2 and TPD54 using insertion at BspEI and BglII, which improved biotinylation (Kim et al., 2016).

## Cell culture

HeLa cells (Health Protection Agency/European Collection of Authenticated Cell Cultures; 93021013) were maintained in DMEM supplemented with 10% FBS and 100 U ml⁻¹ penicillin/streptomycin. RPE1 cells (HD-PAR-541 clone 7724) were maintained in Ham's F12 Nutrient Mixture supplemented with 100 U ml⁻¹ penicillin/streptomycin, 10% FBS, 2.3 g/liter sodium bicarbonate, and 2 mM L-glutamine. A2780 human ovarian cancer cells (female) stably expressing Rab25 (Caswell et al., 2007) were maintained in RPMI-1640 medium (Sigma-Aldrich) supplemented with 10% FCS, 1% L-glutamine, and 1% antibiotic-antimycotic (Sigma-Aldrich). All cell lines were kept at 37°C and 5% $CO_2$. RNAi was done by transfecting 100 nM siRNA (TPD54#1: 5′-GUCCUACCUGUUACGCAAU-3′, TPD54#2: 5′-CUCACGUUUGUAGAUGAAA-3′, TPD54#3: 5′-CAUGUUAGCCCAUCAGAAU-3′; TPD52: 5′-CAAAUAGUUUGUGGGUUAA-3′; TPD53: 5′-GUCUCCAGCAAUAGGAUGAUUUACUA-3′) with Lipofectamine 2000 (Thermo Fisher Scientific) according to the manufacturer's protocol. For DNA plasmids, cells were transfected with a total of 600 ng DNA (per well of a 4-well LabTek dish) using 0.75 µl GeneJuice (Merck Millipore) following the manufacturer's protocol. The A2780 cells were transfected by electroporation using a nucleofector (Lonza; Amaxa) using solution T, program A-23, 20 nM siRNA as per the manufacturer's instructions. Invasion experiments were performed 24 h after nucleofection.

## Protein expression and purification

GST or GST-TPD54 constructs were expressed in *Escherichia coli* BL21 cells grown in double yeast tryptone media. Starter cultures of 10 ml were grown overnight at 37°C and shaken at 200 rpm. They were then diluted into 400-ml cultures and grown at 37°C, 200 rpm until an optical density at 600 nm between 0.6 and 0.8. To induce expression of the proteins, IPTG was added to

a final concentration of 0.5 mM and cells were grown for a further 5 h at 37°C, 200 rpm. The cells were harvested by centrifugation at 9,200 *g* for 10 min at 4°C, washed with cold PBS and pelleted again at 3,200 *g* for 15 min at 4°C. Pellets were stored at –80°C until purification.

For purification, pellets were resuspended in 50 ml lysis buffer (50 mM Tris, 150 mM NaCl, protease inhibitor cocktail tablet [Roche], and 0.2 mM PMSF, pH 8) and lysed by sonication. Cell debris was pelleted by centrifugation at 34,600 *g* for 30 min at 4°C. The supernatant was loaded onto a GSTrap column (GE Healthcare), which was then washed with 10 column volumes of wash buffer (50 mM Tris, 150 mM NaCl, protease inhibitor cocktail tablet, and 0.1 mM PMSF, pH 8) and then 10 column volumes of high-salt wash buffer (50 mM Tris, 500 mM NaCl, protease inhibitor cocktail tablet, and 0.1 mM PMSF, pH 8). The GST-tagged proteins were eluted by the addition of elution buffer (50 mM Tris, 150 mM NaCl, and 50 mM glutathione, pH 8). Purified proteins were dialyzed into binding buffer (150 mM NaCl and 20 mM Hepes, pH 7) for use in liposome-binding assays.

## Protein–liposome interactions

Folch extract from bovine brain (Sigma-Aldrich) was dissolved in chloroform. The lipid mixture was dried with nitrogen flow followed by 2-h desiccation in a vacuum. The dried lipids were resuspended in binding buffer (150 mM NaCl and 20 mM Hepes, pH 7) to a final concentration of 4 mM. To generate liposomes with desired diameters, the lipid mixture was heated to 60°C and extruded 15× through polycarbonate membrane filters (Avanti Polar Lipids) with pore sizes of 200, 100, 50, or 30 nm. For 30-nm and 50-nm sizes, liposomes were produced by first extruding through a 100-nm filter, followed by extrusion through filters of a smaller pore size. Note that material is lost with each extrusion. Liposomes were stored at 4°C until used in binding assays. For the liposome pelleting assay, liposomes were centrifuged at 100,000 *g* for 25 min at 4°C. Pellet and supernatant samples were prepared with NuPAGE LDS Sample Buffer + reducing agent (Invitrogen) and incubated at 70°C for 10 min. Lipids were resolved on 12% Bis-Tris gels run in MES buffer and stained with 0.1% Coomassie in 10% acetic acid for 5 min (method adapted from Boucrot et al., 2012). Gels were destained in water overnight to leach the loading dye.

For the protein–liposome–binding assay, purified GST-tagged proteins were precleared before use, and then protein (2 µM) and liposomes (600 µM) were mixed with binding buffer (150 mM NaCl and 20 mM Hepes, pH 7) to a total volume of 200 µl and incubated on ice for 20 min. Samples were then centrifuged at 100,000 *g* for 25 min at 4°C. Bound (pellet) and free (supernatant) samples were prepared for SDS-PAGE by adding Laemmli buffer and boiling for 5 min. Proteins were resolved by SDS-PAGE on 4% to 15% Mini-PROTEAN TGX gels and visualized by staining with InstantBlue (Expedeon).

To quantify protein–liposome binding, the mean pixel density of gel bands was measured in Fiji. The background for each lane was subtracted, and the precipitation (no vesicle control band) was subtracted from the bound sample bands. These values were normalized to the amount of GST-TPD54 WT bound

to 100 nm liposomes. Liposome pelleting was quantified in the same way, and the amount of lipid pelleted for each liposome size was calculated relative to 100-nm liposome pelleting for each of three experiments. These values were used to correct the protein-binding results.

## Cellular biochemistry

For FLAG immunoprecipitation, HeLa cells were seeded in 10-cm dishes. FLAG-TPD54, FLAG-TPD53, or FLAG-TPD52 was transfected with GFP-FKBP, GFP-FKBP-TPD54, or GFP-FKBP-TPD54 L53P, L67P, 10 µg total using GeneJuice (Merck Millipore) according to the manufacturer's protocol. After 48 h, the cells were lysed with 10 mM Tris, pH 7.5, 150 mM NaCl, 0.5 mM EDTA, 1% Triton X-100, and protease inhibitors. Lysate was passed through a 23-gauge syringe and spun in a benchtop centrifuge for 15 min at 4°C. The cleared lysate was then incubated for 2 h at 4°C with 10 µl anti-FLAG M2 magnetic beads (Sigma-Aldrich; M8823) prewashed with TBS. The beads were then washed three times with cold TBS, resuspended in Laemmli buffer, and run on a precast 4 to 15% polyacrylamide gel (Bio-Rad).

For Western blotting, the following primary antibodies were used: rabbit anti-TPD54 (Dundee Cell Products), 1:1,000; goat anti-TPD53 (Thermo Fisher Scientific; PA5-18798), 0.5 µg/ml; mouse anti-GFP clones 7.1 and 13.1 (Roche; 11814460001), 1:1,000; and mouse anti-FLAG M2 (Sigma-Aldrich; F1804), 1 µg/ml. HRP-conjugated secondary antibodies were used with enhanced chemiluminescence detection reagent (GE Healthcare) for detection, and manual exposure of Hyperfilm (GE Healthcare) was performed.

For GFP immunoprecipitations, three 10-cm dishes of confluent RPE-1 cells transiently expressing either GFP or GFP-TPD54 were used for each condition. Cells were scraped in lysis buffer (10 mM Tris-HCl pH 7.5, 150 mM NaCl, 0.5 mM EDTA, 0.5% NP-40, protease inhibitors [Roche]) and passed through a 23-gauge syringe and then cleared spun in a benchtop centrifuge for 15 min at 4°C. Lysates were incubated for 1 h with GFP-Trap beads (ChromoTek), washed once with exchange buffer (10 mM Tris-HCl pH7.5, 150 mM NaCl, 0.5 mM EDTA) and three times with wash buffer (10 mM Tris-HCl, pH 7.5, 500 mM NaCl, 0.5 mM EDTA). The immunoprecipitations were run on a 4–15% polyacrylamide gel until they were 1 cm into the gel. Before excision and analysis at the University of Dundee Proteomics Facility.

For each BioID experiment, three 15-cm dishes of RPE1 cells per construct were each transfected with 9.5 µg DNA diluted in 1.28 ml Opti-MEM with 72 µl GeneJuice. Next day, media was replaced and supplemented with 50 µM biotin (Sigma-Aldrich). After 16 h, biotin-fed RPE1 cells were recovered and incubated in RIPA buffer (25 mM Tris-HCl, pH 7.6, 150 mM NaCl, 1% NP-40, 0.5% sodium deoxycholate, and 0.1% SDS) for 30 min on ice, followed by 10 passages through a 23-G needle. Cleared lysates were incubated with magnetic streptavidin beads at 4°C for 2 h. Beads were washed with RIPA buffer and then PBS before storage at –20°C. On-bead trypsin digestion was performed before desalting using a C18 stage tip and analysis by nano–liquid chromatography electrospray ionization tandem mass

spectrometry in an Ultimate 3000/Orbitrap Fusion mass spectrometer (Proteomics Research Technology Platform, University of Warwick).

## Integrin recycling assay

The ELISA plate (Thermo Fisher Scientific; Maxisorp 96 wells) was prepared the day before the experiment by incubating the wells with 50 µl/well of 5 µg/ml anti-integrin α5 antibodies (BD Biosciences; 555651) in 0.05 M $Na_2CO_3$, pH 9.6, overnight at 4°C. 10-cm dishes were seeded with RPE1 cells in triplicate. Cells were serum starved for 30 min at 37°C. Following two 5 ml washes with cold PBS, the surface receptors were labeled with 0.133 mg/ml EZ-Link Sulfo-NHS-SS-Biotin (Thermo Fisher Scientific; 21331) in PBS at 4°C for 30 min. Cells were washed twice with 5 ml cold PBS on ice, and 5 ml warm serum-free medium was added. Plates were incubated at 37°C for 30 min to allow receptor internalization and then washed again with 5 ml of cold PBS on ice.

The cells were washed with cold reduction buffer (50 mM Tris, pH 7.5, and 102.5 mM NaCl, pH 8.6). Cell surface was reduced by adding 3 ml reduction buffer and 1 ml Mesna buffer (390 mg Mesna was added to 26 ml reduction buffer and mixed thoroughly, and 39 µl of 10 M NaOH was added). Plates were agitated at 4°C for 20 min and then washed twice with cold PBS. The plates were then incubated in warm medium at 37°C to allow receptor recycling. The cell surface was then reduced again for 20 min as described above. Reduction buffer containing iodoacetamide (442 mg in 26 ml PBS) was added to the reduction buffer (1:4) to quench the reaction, for 10 min. The ELISA plate was blocked with 5% BSA in PBS-Tween at room temperature for 1 h. The cells were washed twice with cold PBS on ice. Lysates were obtained by scraping the cells with a total of 100 µl/condition of lysis buffer (200 mM NaCl, 75 mM Tris, pH 7.5, 15 mM NaF, 1.5 mM $Na_3VO_4$, 7.5 mM EDTA, 7.5 mM EGTA, 1.5% Triton-X100, 0.75% Igepal, and protease inhibitors). The ELISA plate was washed twice with PBS-Tween, and 50 µl lysate was put in each well, covered with parafilm, and incubated overnight at 4°C. Following five washes with PBS-Tween, 50 µl of 1 µg/ml streptavidin-HRP and 1% BSA in PBS-Tween was added to each well and incubated for 1 h at 4°C. Five more washes were performed with PBS-Tween and 50 µl of detection reagent (0.56 mg/ml ortho-phenylenediamine dihydrochloride in ELISA buffer [25.4 mM $NaHPO_4$ and 12.3 mM citric acid, pH 5.4] with 0.003% $H_2O_2$) was added to each well. The plate was incubated at room temperature in the dark for 15 min before reading on a plate reader using 450-nm light.

## Microscopy

For confocal imaging, cells were grown in four-well, glass-bottom, 3.5-cm dishes (Greiner Bio-One), and medium was exchanged for Leibovitz L-15 $CO_2$-independent medium for imaging at 37°C on a spinning disc confocal system (PerkinElmer; Ultraview Vox) with a 100× 1.4 NA oil-immersion objective. Images were captured using an ORCA-R2 digital charge-coupled device camera (Hamamatsu) following excitation with 488-nm and 561-nm lasers. For some experiments a Nikon CSU-W1 spinning disc confocal system with SoRa upgrade (Yokogawa)

was used with a Nikon 100×, 1.49, oil, CFI SR HP Apo total internal reflection fluorescence objective (Nikon) and 95B Prime camera (Photometrics). Rerouting of mCherry-FKBP-TPD52 or mCherry-FKBP-TPD53 to the mitochondria (dark MitoTrap) was induced by addition of 200 nM rapamycin (Alfa Aesar). For the Rab GTPase corerouting experiments, an image before rapamycin and an image 2 min after rapamycin were taken of live cells. For spatiotemporal variance analysis, cells were imaged for 30 frames with a 300-ms exposure.

For widefield imaging of 2D migration, 4-well LabTek dishes were incubated 30 min at 37°C with 10 µg/ml fibronectin or O/N at 37°C with 20 µg/ml laminin (Sigma-Aldrich; L2020-1MG). RPE1 cells were then plated at low density in the dishes and imaged the next day in L15 medium supplemented with 10% FBS and 100 U ml$^{-1}$ penicillin/streptomycin. Cells were imaged at 37°C on a Nikon Ti epifluorescence microscope with a 20× 0.50 NA air objective, a heated chamber (OKOlab) and CoolSnap MYO camera (Photometrics) using NIS Elements AR software. Movies were recorded over 12 h (one frame/10 min or one frame/20 min) with phase contrast. Cells in CDM were imaged for 16 h using an Eclipse Ti inverted microscope (Nikon) with a 20×/0.45 SPlan Fluar objective and the Nikon filter sets for bright field and a pE-300 LED (CoolLED) fluorescent light source with imaging software NIS Elements AR.5.20.02. Images were acquired using a Retiga R6 (Q-Imaging) camera.

For the invasion assay, 5 mg/ml collagen-I supplemented with 25 µg/ml fibronectin was polymerized in inserts (Corning; Transwell) at 37°C for 1 h. The inserts were inverted, and A2780 cells stably expressing Rab25 were seeded on the opposite side of the filter. The inserts were then put in 0.1% serum medium supplemented with 10% FCS, and 100 ng hepatocyte growth factor and 30 ng/ml EGF were added on top of the matrix. After 72 h, cells were stained with Calcein-AM and visualized with an inverted Leica SP8 confocal microscope using a 20× objective. Cells were considered invasive above 45 µm, and slices were taken every 15 µm.

Imaging recycling of immunolabeled integrins was done by incubating RPE-1 cells (control or TPD54 depleted) in serum-free medium for 30 min. Cell surface integrins were labeled on ice for 30 min using either clone VC5 anti-human CD49e (BD PharMingen; 1:500, 555651) or clone SNAKA51 (Sigma-Aldrich; 1:500, MABT201). Following three washes with cold serum-free medium, coverslips were transferred to warm serum-free medium and incubated for 0, 30, or 120 min at 37°C. Finally, cells were fixed in 3% PFA and 4% sucrose in PBS for 15 min at RT, permeabilized with 0.1% Triton X-100 in PBS, and blocked and stained with goat anti-mouse Alexa Fluor 568–conjugated secondary antibodies (1:500 in blocking solution). Coverslips were mounted and imaged by confocal microscopy.

## CLEM

Analysis of vesicle capture and mitochondrial aggregation was by CLEM following the methods outlined previously (Larocque et al., 2020). Briefly, transfected cells were plated onto gridded dishes (MatTek; P35G-1.5-14-CGRD). Cells were imaged at 37°C in Leibovitz L-15 CO$_2$-independent medium supplemented with 10% FBS. Rapamycin (200 nM, final concentration) was added

for 3 min before the cells were fixed in 3% glutaraldehyde and 0.5% paraformaldehyde in 0.05 M phosphate buffer, pH 7.4, for 1 h. Aldehydes were quenched in 50 mM glycine solution and then postfixed in 1% osmium tetroxide and 1.5% potassium ferrocyanide for 1 h and then in 1% tannic acid for 45 min to enhance membrane contrast. Cells were rinsed in 1% sodium sulfate and then twice in H$_2$O before being dehydrated in grade series ethanol and embedded in EPON resin (TAAB). Correlation of light images allowed the cell of interest to be identified for sectioning, and 70-nm ultrathin sections were cut and collected on formvar-coated hexagonal 100-mesh grids (EM resolutions). Sections were poststained with Reynolds lead citrate for 5 min. Electron micrographs were recorded using a JEOL 1400 transmission EM operating at 100 kV using iTEM software.

## Data analysis

Analysis of spatiotemporal variance of fluorescence signals in live-cell movies was done using five 20 × 20–pixel excerpts of 30 frames from GFP-Rab live-cell imaging captured at 0.1775 s per frame. The excerpts were positioned in the cytoplasm away from bright structures. Each frame was first normalized to the mean pixel intensity for that frame and then the variance per pixel over time was calculated, resulting in a 20 × 20 matrix of variances. The mean of the five matrices is presented as the "spatiotemporal variance" for that cell. The analysis of the mCherry-FKBP-TPD54 constructs was measured in the same way, except that one region of interest of 50 × 50 pixels was taken per cell.

Mitochondrial aggregation was measured using a workflow that segmented postrapamycin images and extracted the area and perimeter of objects above threshold. These data were then fed into Igor Pro, where, for objects greater than 0.2 µm$^2$, the compactness of mitochondria was approximated using a circularity formula:

$$f_{circ} = \frac{4\pi A}{P^2},$$

where $A$ is area and $P$ is perimeter. The median circularity per cell was used to compare conditions.

For the Rab screen, corerouting of Rab GTPases was quantified by averaging for each cell, the pixel intensity in the green channel in 10 regions of interest of 10 × 10 pixels on the mitochondria, before and after rapamycin. This mitochondrial intensity ratio ($F_{post}/F_{pre}$) for every Rab was compared with the ratio of GFP in TPD52- or TPD53-rerouted cells. Estimation statistics were used to generate the difference plot shown in Fig. 6. The mean difference is shown together with bias-corrected and accelerated 95% confidence intervals calculated in R using 1 × 10$^5$ bootstrap replications. The heatmap was generated in R (see Data and software availability) using mitochondrial intensity ratio data. Note that when constructing the heatmap, we reanalyzed data for Rab3a and Rab4a from the original TPD54 screen (Larocque et al., 2020), and this has slightly altered the Rab profile for TPD54.

For the 2D migration assay, cells were tracked using the Fiji plugin Manual Tracking by using the center of the nucleus as guide. The data were saved as CSV files and fed into

CellMigration 1.14 in IgorPro for analysis (Royle, 2021). Superplots were used to show experimental effects across experimental repeats (Lord et al., 2020).

Invasion was quantified using the area calculator plugin in Fiji, measuring the fluorescence intensity of cells invading 45 µm or more and expressing this as a percentage of the fluorescence intensity of all cells within the plug. The data were normalized to siCtrl to show relative invasion.

For the cell shape analysis, a scientist blind to the experimental conditions drew with a stylus the outline of each cell in a frame halfway through the migration movies. The coordinates of all cell contours were fed into CellShape 1.01 in IgorPro for analysis (Royle, 2020).

Figures were made with Fiji or Igor Pro 9 (WaveMetrics) and assembled using Adobe Illustrator. Null-hypothesis statistical tests were as described in the figure legends.

### Online supplemental material

Fig. S1 shows the amplification of TPD52-like proteins in cancer. Fig. S2 shows the effect of TPD54 depletion on migration of RPE1 cells on laminin. Fig. S3 shows the effect of TPD54 depletion on the migratory behavior of RPE1 cells on fibronectin. Fig. S4 shows the effect of TPD52 or TPD53 depletion on migration of RPE1 cells. Fig. S5 shows the effect of TPD54 depletion on the shape of migrating RPE1 cells on fibronectin. Video 1 demonstrates spatiotemporal variance of a selection of TPD54 mutants. Video 2 shows TPD52-like proteins on subresolution vesicles. Video 3 compares the spatiotemporal variance of GFP-tagged Rab GTPases. Video 4 shows the effect of TPD54 depletion on RPE1 cell morphology and migration. Video 5 shows the effect of TPD54 depletion on migration of A2780 cells in CDM.

### Data availability

The two software packages that are described in this paper, CellMigration (Royle, 2021) and CellShape (Royle, 2020), are freely available. All code that is specific to this paper is available at https://github.com/quantixed/p054p031.

## Acknowledgments

The authors thank Laura Cooper, Erick Martins Ratamero, and Claire Mitchell of CAMDU (Computing and Advanced Microscopy Unit) for their support and assistance in this work. We would like to thank Francis Barr (University of Oxford, UK) for reagents and Darius Koester for advice and assistance with liposome production. We would also like to thank Miguel Hernández González and Joseph Cockburn for valuable discussion.

This work was supported by the UK Medical Research Council (MR/P018947/1). G. Larocque was supported by Fonds de Recherche du Québec (Nature et technologies) and University of Warwick Chancellor's Award. D.J. Moore was funded by the Medical Research Council Doctoral Training Partnership (MR/N014294/1).

The authors declare no competing financial interests.

Author contributions: G. Larocque performed experimental work, analyzed data, and wrote the paper. D.J. Moore did the liposome-binding assays and BioID2 experiments and helped with the Rab screens. M. Sittewelle carried out additional migration and immunofluorescence experiments. C. Kuey extended the spatiotemporal variance and mitochondrial aggregation analyses. P.J. La-Borde performed the original liposome-binding assays. B.J. Wilson and P.T. Caswell performed the original invasion assays, which were extended by J.H.R. Hetmanski. N.I. Clarke did the vesicle capture CLEM experiments. S.J. Royle analyzed data, wrote computer code, and wrote the paper.

Submitted: 4 September 2020

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

# Supplemental material

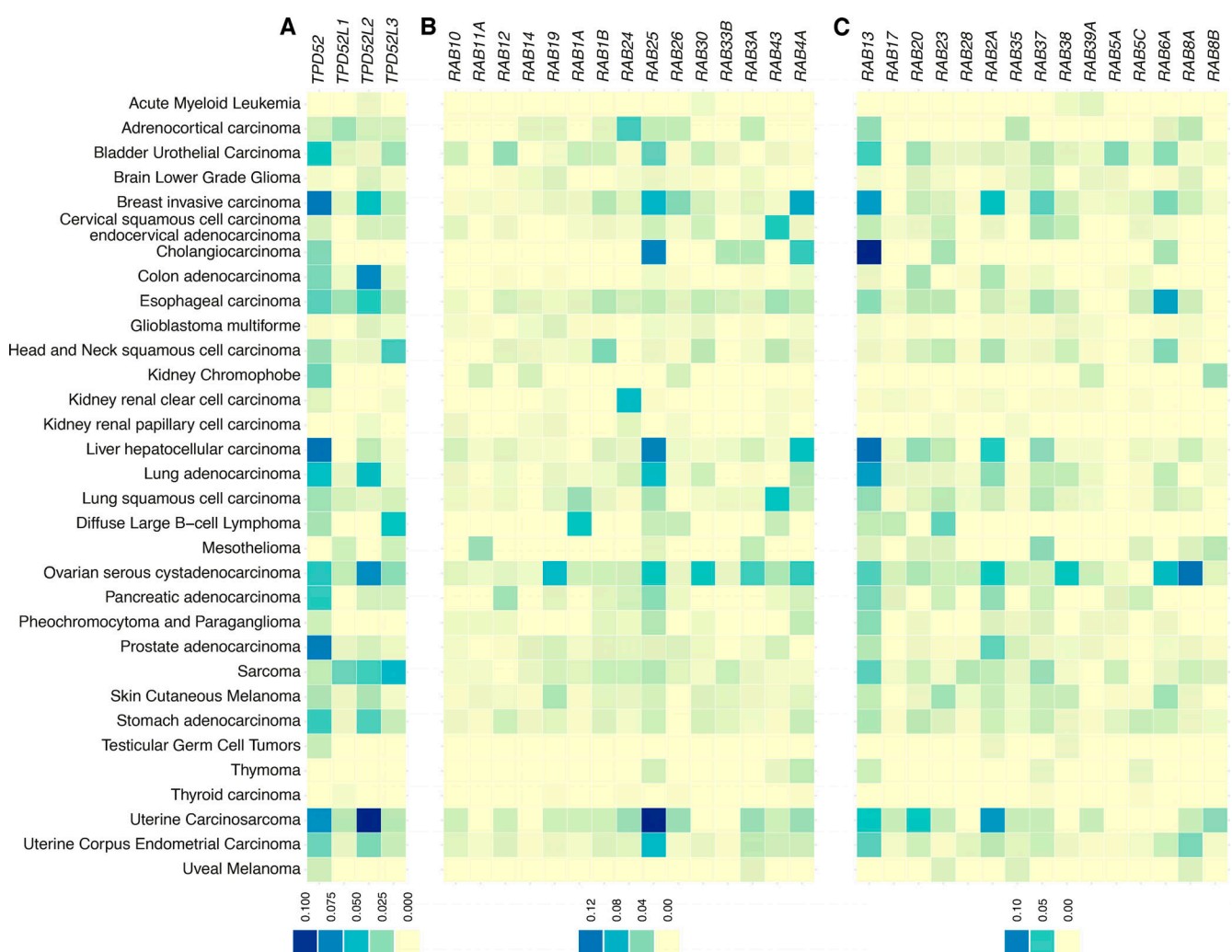

Figure S1. **Amplification of TPD52-like proteins in cancer.** Frequency of amplification of the indicated genes in The Cancer Genome Atlas PanCancer Atlas Studies (10,967 samples). **(A)** TPD52-like proteins. **(B)** Rab GTPases associated with INVs. **(C)** Rab GTPases not associated with INVs. Color scale indicates the fraction of samples of each cancer type showing amplification.

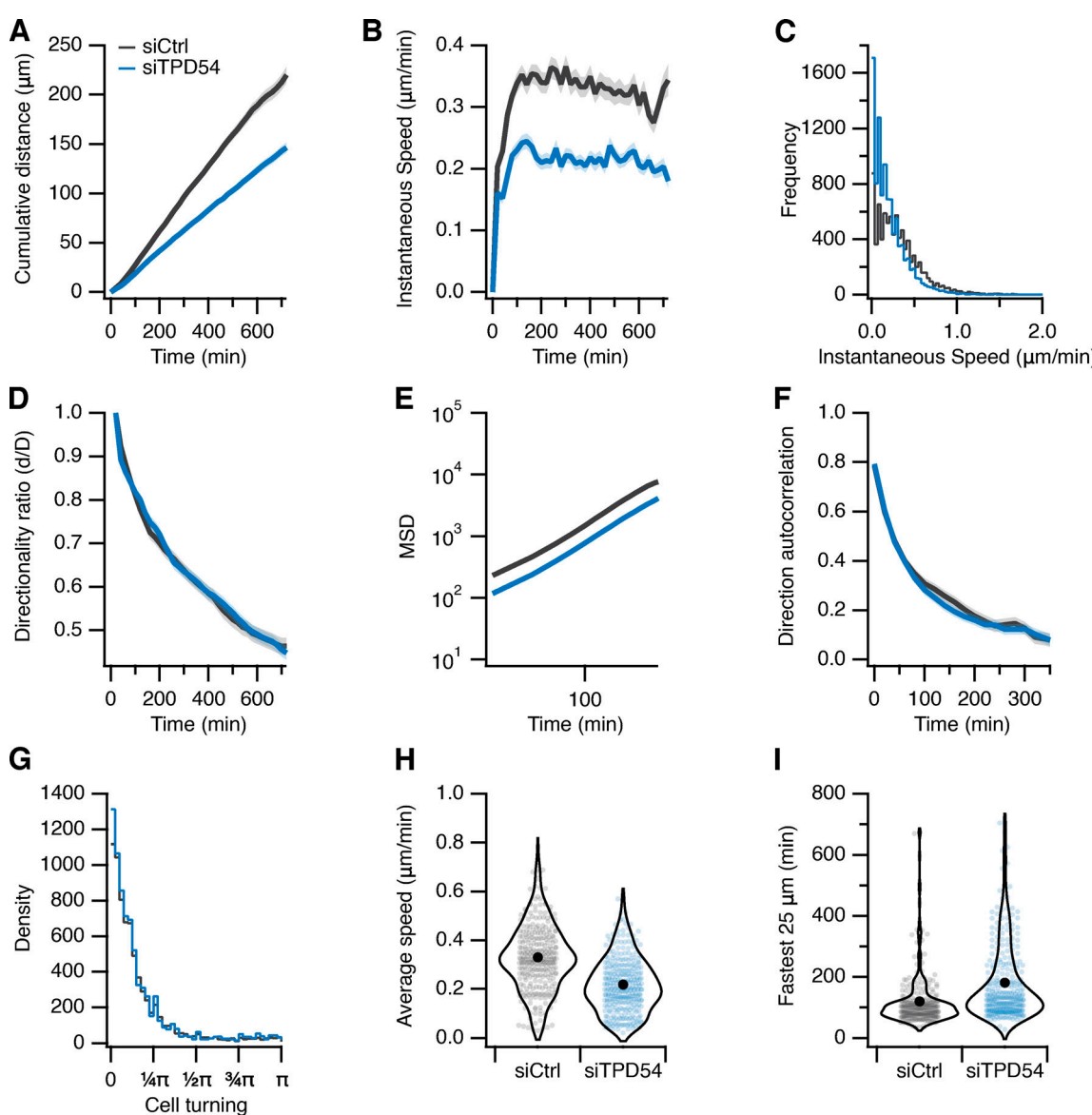

Figure S2.  **Migration of RPE1 cells on laminin. Control (gray) or TPD54-depleted (blue) RPE1 cells migrating on laminin-coated dishes.** $n_{cell}$ = 286–297, $n_{exp}$ = 2. **(A)** Cumulative distance over time (mean ± SEM). **(B)** Instantaneous speed over time (mean ± SEM). **(C)** Histogram of instantaneous speed. **(D)** Directionality ratio over time (mean ± SEM). **(E)** Mean squared displacement (MSD; mean ± SEM). **(F)** Direction autocorrelation (mean ± SEM). **(G)** Histogram of cell turning. The angle distribution for all tracks in the analysis measured from one trajectory to the next. **(H)** Violin plot of average speed of each cell. **(I)** Violin plot of the fastest time taken for each cell to traverse a "segment" of 25 µm in a track. Dots represent individual cells, and marker indicates the mean.

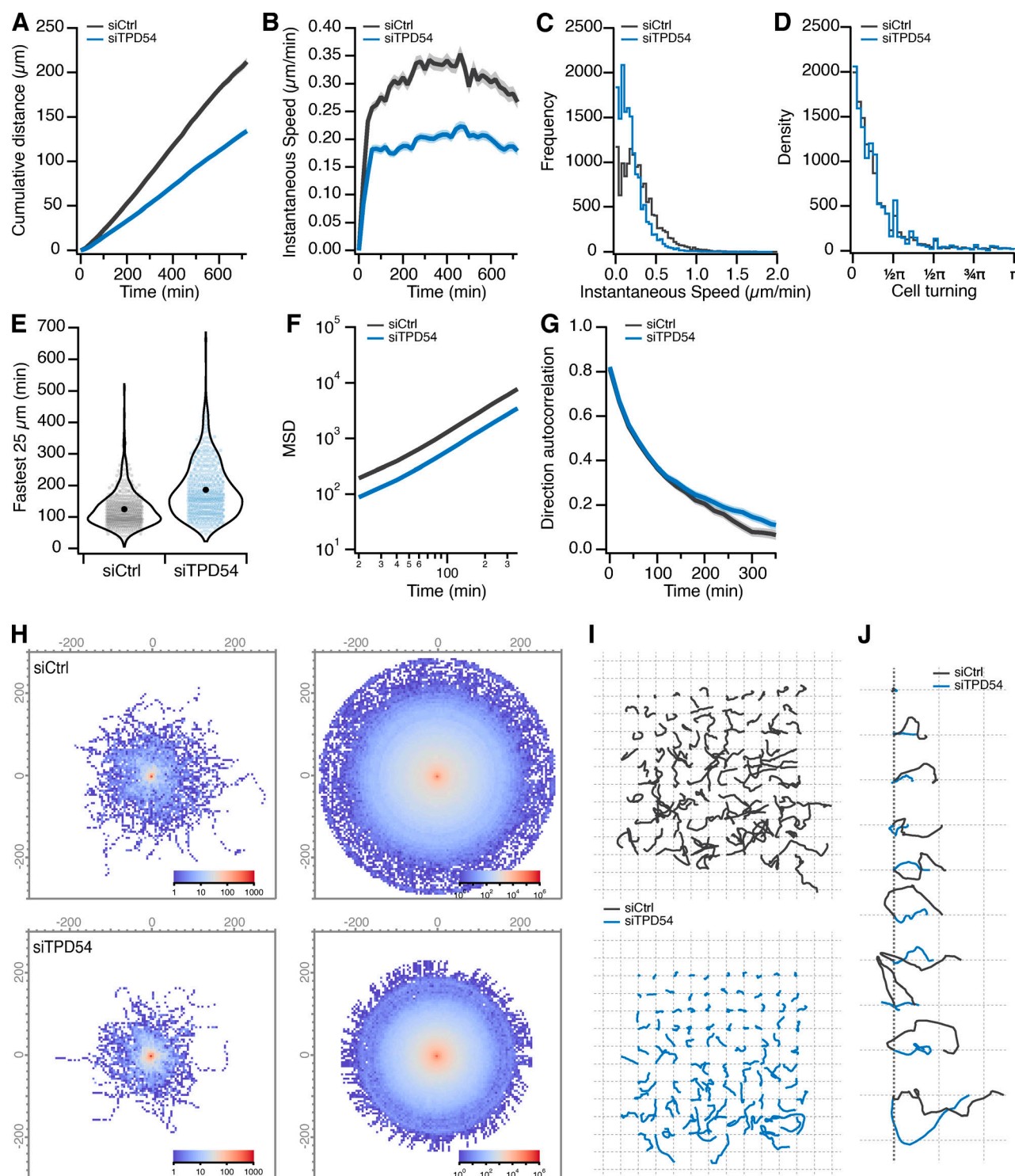

Figure S3. **Effect of TPD54 depletion on migration of RPE1 cells.** Statistics of control (gray) or TPD54-depleted (blue) RPE1 cells migrating on fibronectin-coated dishes. Data from Fig. 8. **(A)** Cumulative distance over time (mean ± SEM). **(B)** Instantaneous speed over time (mean ± SEM). **(C)** Histogram of instantaneous speed. **(D)** Histogram of cell turning. The angle distribution for all tracks in the analysis measured from one trajectory to the next. **(E)** Violin plot of the fastest time taken for each cell to traverse a segment of 25 μm in a track. Dots represent individual cells, marker indicates the mean. **(F)** Mean squared displacement (mean ± SEM). **(G)** Direction autocorrelation (mean ± SEM). **(H)** Overlay of all tracks in the dataset shown as a heatmap (left), bootstrapped and rotated view of cell tracks to visualize the average explored space by cells in the dataset. Density is shown by the log color scale indicated. **(I)** Image quilt of a sample of cell tracks from the dataset. Each track is shown in its original orientation, arrayed on a grid from the shortest distance traveled to the longest. **(J)** Sparkline image of a diagonal sample through the image quilt (I). To visualize directionality, tracks are rotated such that the end point of the track due east from the start.

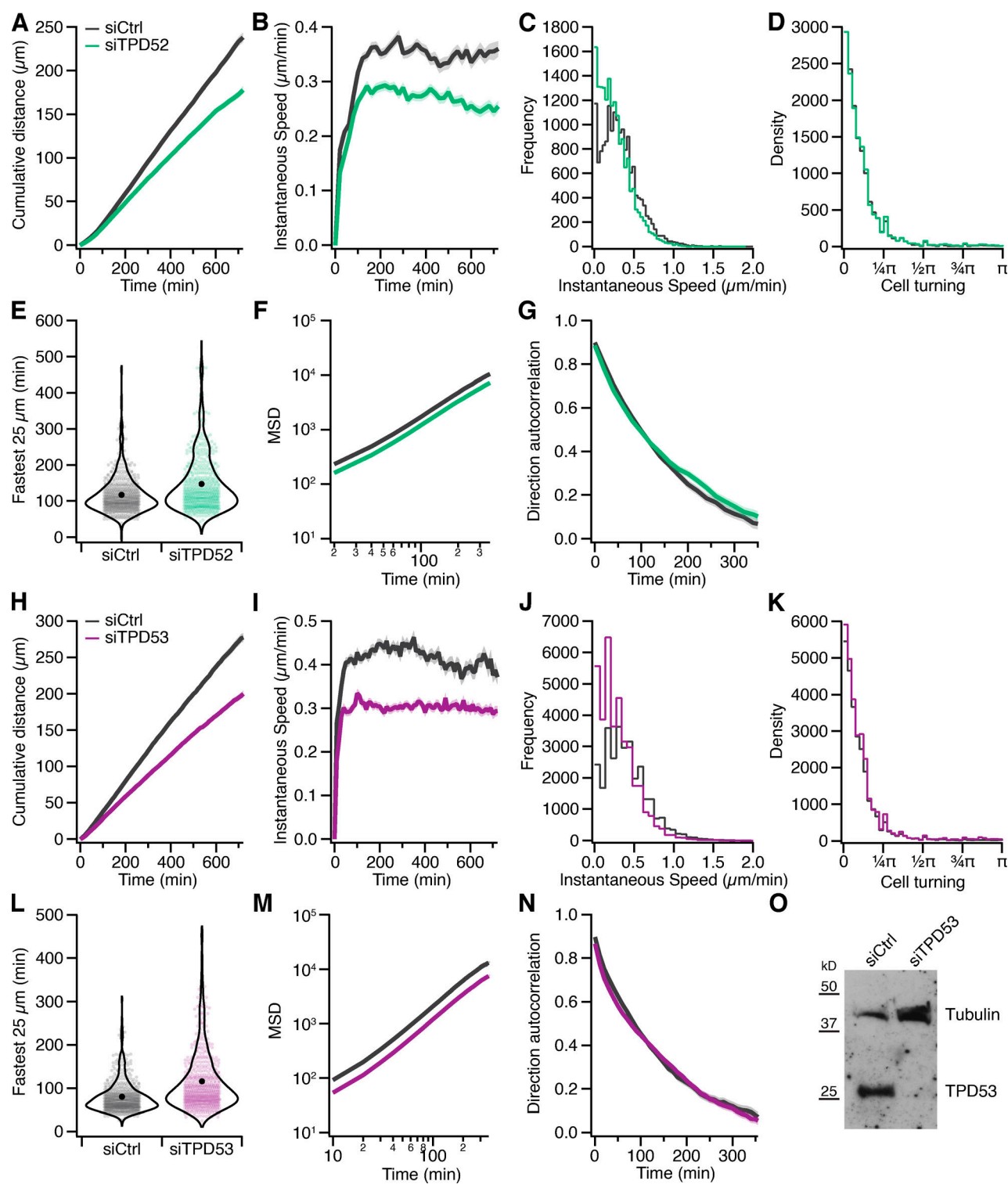

Figure S4. **Effect of TPD52 or TPD53 depletion on migration of RPE1 cells.** Statistics of control (gray), TPD52-depleted (green), or TPD53-depleted (purple) RPE1 cells migrating on fibronectin-coated dishes. Data from Fig. 8. **(A and H)** Cumulative distance over time (mean ± SEM). **(B and I)** Instantaneous speed over time (mean ± SEM). **(C and J)** Histogram of instantaneous speed. **(D and K)** Histogram of cell turning. The angle distribution for all tracks in the analysis measured from one trajectory to the next. **(E and L)** Violin plot of the fastest time taken for each cell to traverse a segment of 25 µm in a track. Dots represent individual cells, and marker indicates the mean. **(F and M)** Mean squared displacement (mean ± SEM). **(G and N)** Direction autocorrelation (mean ± SEM). **(O)** Western blot to show depletion of TPD53 under the conditions of the migration experiment. Depletion of TPD52 was not possible to assess due to lack of specific antibodies.

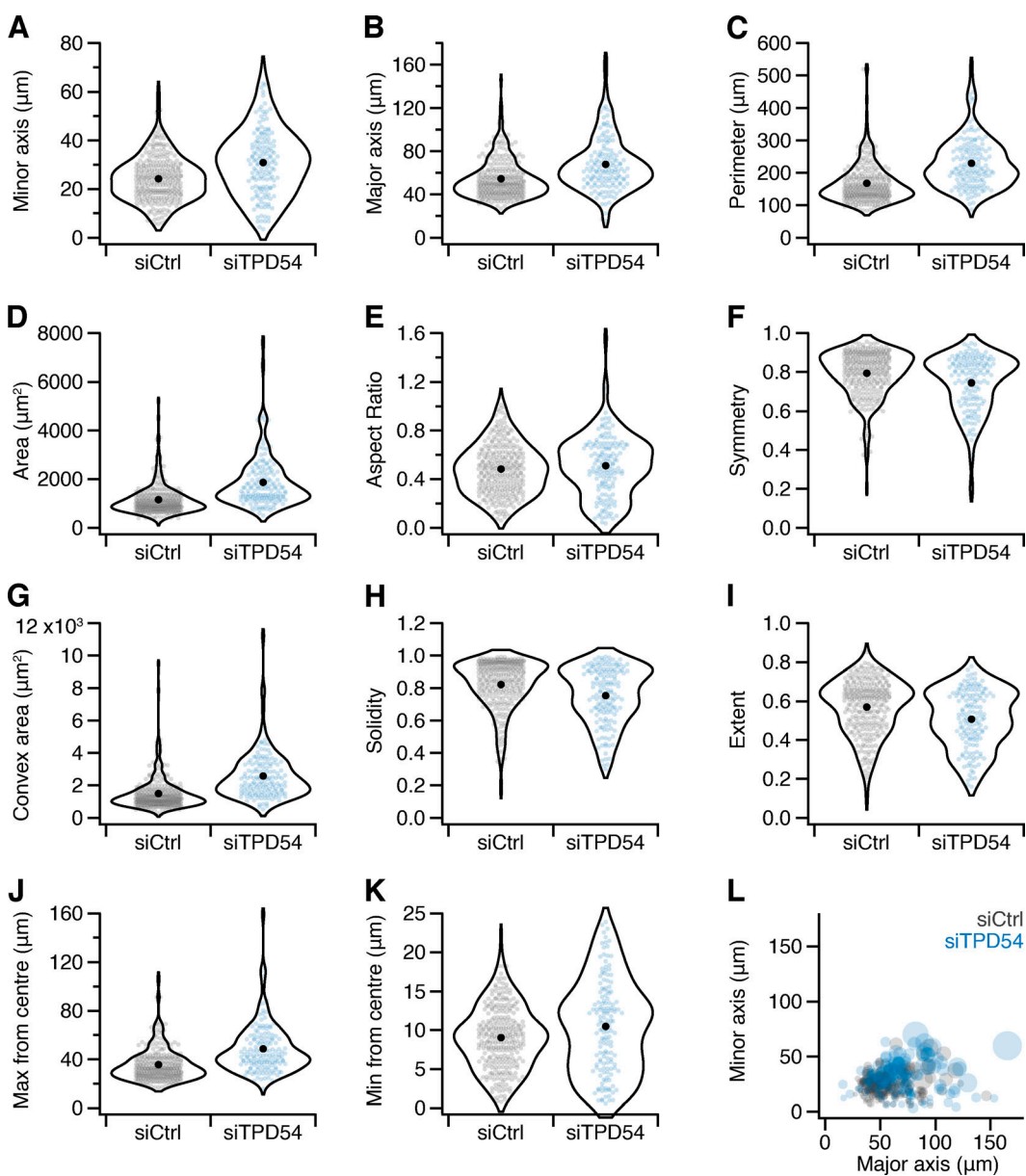

**Figure S5. Effect of TPD54 depletion on the shape of RPE1 cells.** Cell shape descriptors for control (gray) or TPD54-depleted (blue) RPE1 cells on fibronectin-coated dishes. Valid shapes were analyzed from $n_{cell}$ = 288 (siCtrl) and $n_{cell}$ = 151 (siTPD54). **(A–K)** Violin plots where dots represent individual cells and marker indicates the mean, showing minor axis length (A); major axis length (B); cell perimeter (C); cell area (D); aspect ratio, minor/major axes (E); symmetry, ratio of the cell area to the area of the cell footprint reflected on its major axis (F); convex area, area of a convex hull enclosing the cell perimeter (G); solidity, ratio of cell area to convex area (H); extent, ratio of cell area to the area of a bounding box (I); maximum distance to the perimeter from the cell center (J); and minimum distance to the perimeter from the cell center (K). **(L)** A plot of minor axis length as a function of major axis length. Each cell in the dataset is represented by a bubble, the size of which corresponds to cell area.

Video 1. **Comparison of fluorescence variance of TPD54 mutants.** Representative movies of HeLa cells expressing mCherry-FKBP or mCherry-FKBP-TPD54 constructs, as indicated. Movies were captured at 3.33 Hz; playback is 10 Hz. Boxed regions are shown below zoomed to 18×. Scale bars, 10 μm (1×) and 1 μm (18×).

Video 2. **Imaging of TPD52-like proteins on subresolution vesicles.** Live-cell confocal movies of HeLa cells expressing GFP, TPD54, TPD53, or TPD52, captured at ~8 Hz, playback 10 Hz. Scale bar, 1 μm.

Video 3.   **Comparison of fluorescence variance of Rab GTPases.** Representative movies of HeLa cells expressing GFP, GFP-TPD54 or GFP-Rabs, captured at 3.33 Hz, playback is 10 Hz. Boxed regions are shown below zoomed to 10×. Scale bars, 5 µm (1×) and 1 µm (10×).

Video 4.   **Effect of TPD54 depletion on RPE1 cell morphology and migration.** Excerpts of a 12 h recording of RPE1 cells (siControl, left; siTPD54, right) migrating on fibronectin. Playback is 20 Hz. Scale bar, 100 µm. Time, hours:minutes.

Video 5.   **Effect of TPD54 depletion on migration of A2780 cells in CDM.** A 16-h recording of RPE1 cells (siControl, left; siTPD54, right) migrating in CDM. Tracks of individual cells are shown in color. Playback is 10 Hz. Scale bar, 100 µm. Time, hours:minutes.

