## [Peer Review File · The Journal of Cell Biology]

Intracellular nanovesicles mediate $\alpha 5\beta 1$ integrin trafficking during cell migration

Gabrielle Larocque, Daniel Moore, Méghane Sittewelle, Cansu Küey, Joseph Hetmanski, Penelope La-Borde, Beverley Wilson, Nicholas Clarke, Patrick Caswell, and Stephen Royle

Corresponding Author(s): Stephen Royle, University of Warwick

Review Timeline:

Submission Date:	2020-09-04
Editorial Decision:	2020-10-09
Revision Received:	2021-05-29
Editorial Decision:	2021-06-28
Revision Received:	2021-06-30

Monitoring Editor: Johanna Ivaska

Scientific Editor: Andrea Marat

Transaction Report:

DOI: <https://doi.org/10.1083/jcb.202009028>

October 8, 2020

Re: JCB manuscript #202009028

Prof. Stephen J Royle
University of Warwick
Centre for Mechanochemical Cell Biology Division of Biomedical Cell Biology Warwick Medical School
Gibbet Hill Road
Coventry CV4 7AL
United Kingdom

Dear Prof. Royle,

Thank you for submitting your manuscript entitled "Intracellular nanovesicles mediate integrin trafficking during cell migration". The manuscript was assessed by expert reviewers, whose comments are appended to this letter. We invite you to submit a revision if you can address the reviewers' key concerns, as outlined here.

You will see that the reviewers agree that this study is an interesting follow up to your initial paper reporting the discovery of INVs. However, they find that a more thorough analysis of the role of INVs in integrin trafficking is necessary to ensure that your main conclusion that INVs mediate integrin trafficking during cell migration is sufficiently supported by the data. We agree that as this work will lay the foundation for future mechanistic insight, it is imperative that the descriptive analysis is expanded, for example to include an analysis of the recycling of other integrins and to analyze the presence of integrins in INVs. In addition, we hope that you will be able to address all of the remaining reviewer comments in your revised manuscript.

GENERAL GUIDELINES:

Text limits: Character count for an Article is < 40,000, not including spaces. Count includes title page, abstract, introduction, results, discussion, acknowledgments, and figure legends. Count does not include materials and methods, references, tables, or supplemental legends.

Figures: Articles may have up to 10 main text figures. Figures must be prepared according to the policies outlined in our Instructions to Authors, under Data Presentation, <https://jcb.rupress.org/site/misc/ifora.xhtml>. All figures in accepted manuscripts will be screened prior to publication.

IMPORTANT: It is JCB policy that if requested, original data images must be made available. Failure to provide original images upon request will result in unavoidable delays in publication. Please ensure that you have access to all original microscopy and blot data images before submitting your revision.

Supplemental information: There are strict limits on the allowable amount of supplemental data. Articles may have up to 5 supplemental figures. Up to 10 supplemental videos or flash animations are allowed. A summary of all supplemental material should appear at the end of the Materials and methods section.

As you may know, the typical timeframe for revisions is three to four months. However, we at JCB realize that the implementation of social distancing and shelter in place measures that limit spread of COVID-19 also pose challenges to scientific researchers. Lab closures especially are preventing scientists from conducting experiments to further their research. Therefore, JCB has waived the revision time limit. We recommend that you reach out to the editors once your lab has reopened to decide on an appropriate time frame for resubmission. Please note that papers are generally considered through only one revision cycle, so any revised manuscript will likely be either accepted or rejected.

Thank you for this interesting contribution to Journal of Cell Biology. You can contact us at the journal office with any questions, cellbio@rockefeller.edu or call (212) 327-8588.

Sincerely,

Johanna Ivaska
Monitoring Editor

Andrea L. Marat
Senior Scientific Editor

Journal of Cell Biology

Reviewer #1 (Comments to the Authors (Required)):

Ms. No. 202009028, Journal of Cell Biology

Intracellular nanovesicles (INVs) have recently been discovered by the team of Stephen Royle and his colleagues. Larocque and colleagues have now studied the mode of binding of TPD proteins with INVs, the complement of Rab proteins that are found on INVs, and the effect of INVs on cell migration and integrin recycling. A quite substantial body of data is presented.

The detailed characterization of a novel intracellular organelle like INVs is obviously of interest to a wide readership in cell biology. How these are generated and how they relate to other intracellular organelles in terms of cargo trafficking (if this is indeed their prime role) remain unanswered questions at this stage. In the current manuscript, the interaction of TPD protein family members with INVs has been addressed. The curvature preference for small radii that is described in Fig. 2D is rather partial. This may explain why TPD proteins are not only found on INVs, and also on

endosomes and Golgi.

The R159E mutant was generated to weaken the charge interaction of TPD54 with membranes. The mitochondrial aggregation phenotype that is seen with the rerouting of wild-type TPD54 (Fig. 2E) is visibly lost with this mutant (Fig. 2F). The manuscript would clearly be strengthened if this phenotype could be quantified.

On page 3, right column, it is written that "These results confirmed that the coiled-coil domain of TPD54 is responsible for homo- and heteromerization and that the L53P,L67P mutation blocks multimerization." Since Fig. 3B shows co-immunoprecipitation experiments (as opposed to experiments on purified proteins), the conclusion on multimerization appears to be a bit too affirmative and may need to be tuned down a bit.

The cell migration effects that are shown in Figures 6B-H are rather quite minor. It would strengthen the current submission if the authors could explicit their ideas on how these small effects reflect on the physiological contribution(s) of INVs in general, and in integrin trafficking in particular.

Minor points:

- Page 4, left column: "... with TPD52 (Figure 4B) and TPD53 4D), ..." needs to be corrected to "... with TPD52 (Figure 4B) and TPD53 (Figure 4D), ...".

Reviewer #2 (Comments to the Authors (Required)):

This work is a follow up of recent report by the same group on the characterization via spatiotemporal fluctuation (variance) of fluorescence microscopy of sub resolution nanovesicles (INV). The INV are characterized by the presence of TPD52-like proteins which represent the unique so far identified markers. The authors begin this work by examining the minimal domain required for the association of INV. Next use a mitochondrial aggregation assays to explores the role of TPD52, 53 and 54 protein on INV and the association with various RABs. They finish by analysing the impact of TPD52-like protein silencing on migration and invasion and on alpha5 integrin trafficking.

The INVs remain a somewhat elusive sub resolution structure. This work is step forward in their characterization and presumptive functional roles. However, it seems more a set of overall well documented, but somewhat marginal advances and it is dubious whether this set of results might be more suitable for more specialized journals. This said the work contains a set of well documented and presented experiments. The most critical shortcoming is related the role of INV in alpha5 integrin recycling. The findings are clear, and these results might account for the altered cell shape and migration. However, they also call for the mechanisms as to how integrin alpha 5 is recycled. The canonical view is that this route relies on recycling vesicles larger in size with respect to the INV and begs the question of the relationship between INV and recycling vesicles. Additionally, it is unclear whether Integrin alpha 5 is found on INV. This would appear a critical aspect to investigate in order to clarify the role of INV in integrin trafficking. For example, is RAB25 down-regulation impacting also on INV? Does the down-regulation of TPD54 and RAB25 enhance the impact on recycling and migration?

There are also a set of points that requires clarifications or additional experiments.

1. The mutant R159E, which display a reduced spatiotemporal variance, retains a significant extent of binding to liposomes of different size. To reinforce this finding, other mutants and/or fragments of TPD54 should be tested, including the K175E, K177E and the 1-82 fragments.

2. In figure 2E and whenever the mitochondria aggregation assay is used, there would be the need to quantify the extent of aggregation.
3. In figure 3, to reinforce the finding that other family members of the TPD52-like family, and namely TPD52, directly associate to INV, the authors should test them in a mitochondria aggregation assay after KD of TPD54.
4. The finding that RAB18, 9A AND 15 display an apparent high Spatiotemporal variance, but they are not associated to TPD52-like proteins in mitochondria aggregation assays is explained as a potential false positive result due to the association to structures larger than INVs. This is a plausible explanation, but calls for a more precise definition as to the criteria as to when spatiotemporal variance can be interpreted as indication of binding to INV as opposed to larger structures.
5. The analysis of TCGA Pan Cancer Atlas pointing to the amplification of various TPD52-like proteins should be interpreted with caution. Amplification is generally deduced from changes of CNV and its real extent is not directly determined. In addition, the meaning of this CNV variation remains unclear and very weakly correlative of a possible role of these proteins in cancer progression.
6. The impact on cell migration of silencing TPD52-like proteins individually is statistically significant but marginal. It would be relevant and more convincing to assess whether the concomitant silencing of TPD52 and 53 has more pronounced effects.
7. The impact of TPD54 on invasion into FN rich matrix appears somewhat more robust. It would be important to rescue this phenotype and test the role of the TPD52 and 53 on the same assay.

Reviewer #3 (Comments to the Authors (Required)):

This manuscript builds on a previous publication by the same lab and in this same journal reporting the discovery of intracellular nanovesicles (INVs) characterized by the presence of TPD54, a member of the TPD52-like protein family. In this study, the authors define molecular determinants of TPD54 association with membranes (a positively charged stretch of amino acids) and homo- and hetero-dimerization (a predicted coiled-coil (CC) sequence). In addition, the authors identify Rab proteins that associate with INVs containing TPD52 and TPD53, as a complement to their previously published identification of Rabs that associate with TPD54-containing INVs. Finally, the authors show that depletion of TPD52-family proteins causes changes in cell shape and migration, and reduces integrin alpha-5 recycling. The experimental evidence and data analyses presented in the paper are of excellent quality. However, the paper reads like a collection of additional data on TPD52-family proteins that are only tenuously connected to the main conclusion of the paper: that INVs mediate integrin trafficking during cell migration. Moreover, this latter conclusion is insufficiently supported by the data. For example, a detailed study of the role of TPD52-family proteins and INVs in integrin trafficking would require analyses of the recycling of other integrins, integrin expression at the cell surface, presence of integrins in INVs, etc. Additional concerns are described below.

Major comments

1. Fig. 1. The detection of INVs in this study is largely based on indirect methods. It requires for the readers to go back to the authors' previous paper for evidence of TPD protein association with INVs. For example, the association of TPD proteins and their variants with INVs is measured by "spatiotemporal variance of fluorescence", but this method is not explained in the current manuscript. In the videos it is not even clear if the dots are individual nanovesicles or nanodomains of larger organelles. The reader might wonder why the INVs and their associated proteins were not

analyzed by STORM, as in the previous study, or biochemically characterized after isolation by subcellular fractionation, as can be done for similarly sized synaptic vesicles.

2. Fig. 1. It would be interesting to determine if the 155-206 sequence in TPD54 is sufficient for recruitment to INVs. In addition, it would be worth mentioning that the 83-206 sequence lacking the CC domain is recruited to membranes, in line with the subsequent finding that multimerization mediated by this domain is dispensable for binding to INVs.

3. The data in Fig. 2D require analysis of statistical significance of differences. Overall, there doesn't seem to be a strong preference of GST-TPD54 for small liposomes or a strong requirement of R159 for association with liposomes of any size. It is hard to understand how these small differences could explain preferential association of TPD proteins with INVs vs. larger organelles, where these proteins are also present.

4. Figs. 6 and 7. Since TPD proteins are overexpressed in some cancers, it would be important to analyze if TPD protein overexpression causes changes in cell shape and migration, and in integrin trafficking; the effects would be expected to be opposite to those caused by TPD protein knockdown. This would complement the bioinformatic analysis (Fig. S3) and provide support to the hypothesis that up-regulation of TPD proteins enhances the metastatic potential of certain cancers.

5. In the cell migration results shown in Figs. 6B, F and H, is there a way to determine the statistical significance of differences? The curves look similar to me, and I don't see how the authors could draw conclusions about differences in directionality. Also, in Figure 6D, I wonder if the authors tried to rescue the TPD54 KD phenotype by overexpressing TPD52 or TPD53, to determine if there is functional redundancy among the members of this family. This could help interpret the low amplification profiles of some of the TPD52-like proteins in the cancer cells analyzed from the PanCancer Atlas.

6. The authors should explain the rationale for investigating the trafficking of integrin alpha-5 and not other integrins. The title of the paper seems to imply a general role of INVs in integrin trafficking. In addition, the authors state "TPD54-depleted cells had a strong reduction in migration speed on both substrates, indicating that this defect is not restricted to a single integrin heterodimer." The trafficking of other integrins should be studied to draw such general conclusions.

7. An in-depth analysis of the role of TPD proteins in integrin trafficking requires measurement of integrin levels at the cell surface and a demonstration that integrins localize to INVs. Could it be that the role of TPD proteins in integrin trafficking is mediated by association with organelles other than INVs?

8. The first sentence of the "Discussion" section should be changed to "...we demonstrated the involvement of TPD52-like proteins in integrin trafficking...", since I don't see direct evidence that INVs are the responsible for that role.

Minor comments:

9. Fig. 3B. It would be helpful to show the input and immunoprecipitated amounts of the FLAG-TPD proteins co-expressed with the GFP-TPD54 proteins.

10. In the "Introduction" section, there is a typo: "In breast cancer, TPD52 overexpression correlates with poor prognosis a decrease in metastasis-free survival". It should read "In breast cancer, TPD52 overexpression correlates with poor prognosis and decrease in metastasis-free survival".

11. Authors should mention in the "Methods" section, or at least in the legend to Fig. 1/S1, how they predicted the presence of coiled coils in these proteins.

12. Page 2. In the sentences "Mutation of either K154, R159, K165 and K175/K177 to glutamic acid reduced the variance to levels similar to control (Figure 1B). Whereas similar mutations (K15E or R95E) outside this region had no significant effect" the period should be changed to a comma.

13. Page 4, there are two typos in the same sentence that starts with "As with TPD54, Rab30 was

- the strongest hit with TPD52 (Figure 4B) and TPD53 (Figure 4D),...". Please, correct with "As with TPD54, Rab30 was the strongest hit with TPD52 (Figure 4B) and TPD53 (Figure 4D),...".
14. Please, define XFP in the legend to Figs. 2E and S2A schemes, or replace it by mCherry (Fig. 2E) and GFP (Fig. S2A).
 15. Legend to Fig. 4. Define Fpost/Fpre or direct the reader to the "Methods" section to find its definition.
 16. Page 6. In the sentence "We found that TPD54-depleted cells had a strong reduction in migration speed on both substrates" say that depletion was by siRNA treatment.
 17. Page 6, section "Amplification of TPD52-like proteins...". The penultimate sentence needs to end with a parenthesis ")".
 18. In the sentence, "We compared cells transfected with siCtrl expressing GFP with those transfected with siTPD54 that re-expressed GFP or GFP-TPD54 WT (Figure 6 D)" explain that the re-expressed construct was siRNA-resistant.
 19. In the "Discussion" section, there's a missing "s" after "heterodimer" in the question: "...how many integrin heterodimer can be trafficked in an INV?".
 20. Also, in the last sentence of the "Discussion" section, when claiming that the "maximum capacity of an INV is surprisingly high", authors may consider mentioning the actual numbers.
 21. Similar to TPD54, indicate the ID (GeneBank, Uniprot) of the isoform used in the cloning of TPD52 and TPD53 constructs.
 22. Indicate the backbone/vector used to obtain the GST-tagged constructs.

Reviewer 1

We thank the reviewer for their time and valuable comments on our manuscript. Revising the manuscript has been difficult for us due to the pandemic, but we have now addressed all the points raised.

Intracellular nanovesicles (INVs) have recently been discovered by the team of Stephen Royle and his colleagues. Larocque and colleagues have now studied the mode of binding of TPD proteins with INVs, the complement of Rab proteins that are found on INVs, and the effect of INVs on cell migration and integrin recycling. A quite substantial body of data is presented.

The detailed characterization of a novel intracellular organelle like INVs is obviously of interest to a wide readership in cell biology. How these are generated and how they relate to other intracellular organelles in terms of cargo trafficking (if this is indeed their prime role) remain unanswered questions at this stage. In the current manuscript, the interaction of TPD protein family members with INVs has been addressed. The curvature preference for small radii that is described in Fig. 2D is rather partial. This may explain why TPD proteins are not only found on INVs, and also on endosomes and Golgi.

This is an excellent point that we have made clear in the discussion. Our revision experiments have revealed that the liposome-binding we observe is due to a region N-terminal to the conserved patch we identified. Both regions are required for association with INVs, and also endosomes and Golgi.

The R159E mutant was generated to weaken the charge interaction of TPD54 with membranes. The mitochondrial aggregation phenotype that is seen with the rerouting of wild-type TPD54 (Fig. 2E) is visibly lost with this mutant (Fig. 2F). The manuscript would clearly be strengthened if this phenotype could be quantified.

We have developed an automated image analysis method to quantify the aggregation of mitochondria and have applied it throughout the manuscript. In the revised paper, we now show all mutants and truncations from Figure 1 using this method in Figure 3; and use it to test whether TPD52 and TPD53 are on INVs independently of TPD54 in Figure 5.

On page 3, right column, it is written that "These results confirmed that the coiled-coil domain of TPD54 is responsible for homo- and heteromerization and that the L53P,L67P mutation blocks multimerization." Since Fig. 3B shows co-immunoprecipitation experiments (as opposed to experiments on purified proteins), the conclusion on multimerization appears to be a bit too affirmative and may need to be tuned down a bit.

Previous work from Jennifer Byrne showed that TPD52 and TPD53 homo and heteromerize directly, although TPD54 itself was not tested. The involvement of a coiled-coil in multimerization makes direct association most likely. While we think that direct association is most likely, the referee is correct that we haven't demonstrated this. Therefore, we have added: *We assume that multimerization is direct between TPD52-like protein monomers, although our experiments do not rule out an intermediary protein.*

The cell migration effects that are shown in Figures 6B-H are rather quite minor. It would strengthen the current submission if the authors could explicit their ideas on how these small effects reflect on the physiological contribution(s) of INVs in general, and in integrin trafficking in particular.

This criticism was very useful for us. The `CellMigration` program we have developed calculates all kinds of statistics and we were keen to show this off, however the main migration phenotype we have is an effect on speed. The differences in direction were indeed minor and we have removed those plots from the main paper (they are still present in the Supplementary Information).

We have made a number of other changes which strengthen aspects relating to TPD52 and TPD53 on

INVs and on the presence of integrins in INVs. The resulting revamped section on cell migration is now stronger and more focussed on the main phenotype we observed.

Reviewer 2

We thank the reviewer for their helpful comments on our paper. We have addressed all of the points raised, many with new experiments that, due to the pandemic, have taken us considerable time and effort to complete.

This work is a follow up of recent report by the same group on the characterization via spatiotemporal fluctuation (variance) of fluorescence microscopy of sub resolution nanovesicles (INV). The INV are characterized by the presence of TPD52-like proteins which represent the unique so far identified markers. The authors begin this work by examining the minimal domain required for the association of INV. Next use a mitochondrial aggregation assays to explores the role of TPD52, 53 and 54 protein on INV and the association with various RABs. They finish by analysing the impact of TPD52-like protein silencing on migration and invasion and on alpha5 integrin trafficking.

The INVs remain a somewhat elusive sub resolution structure. This work is step forward in their characterization and presumptive functional roles. However, it seems more a set of overall well documented, but somewhat marginal advances and it is dubious whether this set of results might be more suitable for more specialized journals. This said the work contains a set of well documented and presented experiments. The most critical shortcoming is related the role of INV in alpha5 integrin recycling. The findings are clear, and these results might account for the altered cell shape and migration. However, they also call for the mechanisms as to how integrin alpha 5 is recycled. The canonical view is that this route relies on recycling vesicles larger in size with respect to the INV and begs the question of the relationship between INV and recycling vesicles. Additionally, it is unclear whether Integrin alpha 5 is found on INV. This would appear a critical aspect to investigate in order to clarify the role of INV in integrin trafficking. For example, is RAB25 down-regulation impacting also on INV? Does the down-regulation of TPD54 and RAB25 enhance the impact on recycling and migration?

Our model for INVs in receptor recycling is that they operate in addition to (or as subdivisions of) larger recycling vesicles. We are not challenging the canonical model of receptor recycling. In our original submission, we lacked evidence for $\alpha 5$ integrin in INVs and also generalized our conclusion that INVs recycle integrins. We have now focussed on $\alpha 5\beta 1$ integrin recycling via INVs and have changed the title of the manuscript accordingly.

We have added new proteomic data that shows a number of integrin types in INVs. First, an IP of TPD54-positive membrane structures that shows TPD52-like proteins, Rabs and integrins. Second, a BioID analysis of WT vs R159E (cannot localize to INVs) which identified a smaller set of Rabs and integrins. We also added imaging of $\alpha 5$ integrin uptake and recycling in control and TPD54-depleted cells to complement the ELISA-based assay. Although several integrin types were identified in the proteomic analyses and a migration phenotype was also observed on laminin (implicating $\alpha 3$ or $\alpha 6$), our manuscript only assesses recycling of $\alpha 5\beta 1$.

The questions about Rab25 are interesting but we don't have a way to tackle them. It is difficult to examine the impact of Rab25 on INVs because although we can use spatiotemporal variance and mitochondrial aggregation, we know from previous work that taking out one Rab (out of the >16 Rabs on INVs) doesn't have much of an effect on the overall population of INVs. Secondly, the expression of Rab25 in the cell lines we used is quite low. We have tested knockdown of Rab11a together with TPD54 and found no additional effect on migration speed but compensation by Rab11b potentially confounds this result. The invasion of A2780 cells that stably express Rab25 are our best link between TPDs, INVs and Rab25.

There are also a set of points that requires clarifications or additional experiments.

1. The mutant R159E, which display a reduced spatiotemporal variance, retains a significant extent of binding to liposomes of different size. To reinforce this finding, other mutants and/or fragments of TPD54 should be tested, including the K175E, K177E and the 1-82 fragments.

In the revised paper we conclude that the liposome-binding experiments demonstrate direct membrane binding but do not fully recapitulate the association with INVs. The Reviewer is correct that it was puzzling that R159E, which has such a profound effect in cells, did not reduce liposome binding. We have now tested a number of additional mutants in this assay: 1-82, 1-155, K154E, K175,177E, K165E. With the exception of 1-82, all showed liposome binding (Figure 2). Given this finding, we tested TPD54 mutants in a different INV-association assay (mitochondrial aggregation after INV capture). Figure 3 shows that broadly this assay agrees with the spatiotemporal variance and suggests that the liposome binding experiments do not report on INV association, only on direct membrane binding. These experiments have clarified how TPD52-like proteins associated with INVs: due to a membrane binding domain between 83-155 (likely 83-125 and via positive charge in the C-terminus). Thanks to additional truncations suggested by the other referees we are also able to conclude that both of these molecular properties are necessary but neither are sufficient by themselves.

2. In figure 2E and whenever the mitochondria aggregation assay is used, there would be the need to quantify the extent of aggregation.

As described above, the mitochondrial aggregation assay is now more prominent in the manuscript. It now features in Figures 3 and 5 and it has been quantified using an automated image analysis workflow.

3. In figure 3, to reinforce the finding that other family members of the TPD52-like family, and namely TPD52, directly associate to INV, the authors should test them in a mitochondria aggregation assay after KD of TPD54.

This was a great suggestion and we have added this analysis in a new Figure 5. We see aggregation following rerouting of GFP-FKBP-TPD52, -TPD53 and -TPD54, in control cells or when the endogenous TPD54 has been depleted. TPD53 has the weakest effect of all three TPDs, which is a theme that recurs in the paper (fewer hits in Rab screen, weak effect in 2D migration, no effect in invasion). These experiments were therefore very useful to tie together other observations in the paper.

4. The finding that RAB18, 9A AND 15 display an apparent high Spatiotemporal variance, but they are not associated to TPD52-like proteins in mitochondria aggregation assays is explained as a potential false positive result due to the association to structures larger than INVs. This is a plausible explanation, but calls for a more precise definition as to the criteria as to when spatiotemporal variance can be interpreted as indication of binding to INV as opposed to larger structures.

There is a false discovery rate with all screens and we're happy that this explains the results for these Rabs in particular. However, in response to this comment, we spent some time trying to refine the spatiotemporal variance method and were unable to optimize it further or introduce a better definition of what value corresponds to INV-binding. The method is useful and the work on Rabs shows that it has broader applicability than only for TPD52-like proteins, but it is important to verify proteins with high variance in another assay such as mitochondrial aggregation. This is the general strategy in the paper and the concordance looks good.

5. The analysis of TCGA Pan Cancer Atlas pointing to the amplification of various TPD52-like proteins should be interpreted with caution. Amplification is generally deduced from changes of CNV and its real extent is not directly determined. In addition, the meaning of this CNV variation remains unclear and very weakly correlative of a possible role of these proteins in cancer progression.

This was a very useful comment. We agree that this analysis by itself is not conclusive. In the paper we cite work by Jennifer Byrne's group on Tumor Protein D52-like proteins and cancer as well as literature on Rab25, integrins and cancer. Together they make a reasonable case for pursuing the role of TPD52-like proteins in cancer and cell migration. We don't feel that the TCGA Pan Cancer Atlas analysis is given undue prominence.

6. The impact on cell migration of silencing TPD52-like proteins individually is statistically significant but marginal. It would be relevant and more convincing to assess whether the concomitant silencing of TPD52 and 53 has more pronounced effects.

Thank you for this suggestion. In the manuscript we present data on co-depletion of TPD52 or TPD53 with TPD54 in 2D migration in RPE-1 cells on fibronectin. We see no evidence for enhancement of phenotype with co-depletion. It seems likely that TPD54 is the most abundant TPD52-like protein in RPE-1 cells, followed by TPD52 and TPD53. In response to Reviewer 3 (point 4) we have now measured 2D migration following overexpression of TPD52 and TPD53. Conversely, we see a modest enhancement of migration with overexpression of either protein, which is in agreement with the knockdown data.

7. The impact of TPD54 on invasion into FN rich matrix appears somewhat more robust. It would be important to rescue this phenotype and test the role of the TPD52 and 53 on the same assay.

This part of the manuscript has been expanded in two ways. First, we have now also measured migration of A2780 cells stably expressing Rab25 within cell-derived matrix (again FN-rich) as a second measure of invasion. Second, we also depleted TPD52 or TPD53 in addition to depleting TPD54 in both invasion assays. We find that depletion of TPD52 has a similar phenotype to TPD54 but we saw no defect in invasion with depletion of TPD53. We verified that there is some weak expression of TPD53 in these cells and that the siRNA was effective, but presumably TPD54 or TPD52 can compensate for its loss. It seems likely that TPD53 is just not as well targeted to INVs or those INVs involved in integrin recycling specifically (since fewer Rabs were found in the TPD53 screen and Rab11a and Rab25 were negative).

Reviewer 3

We are very grateful to the reviewer for their time and for their constructive comments about our manuscript. It has taken us some time due to the pandemic, but we now have an answer for all of the points raised below.

This manuscript builds on a previous publication by the same lab and in this same journal reporting the discovery of intracellular nanovesicles (INVs) characterized by the presence of TPD54, a member of the TPD52-like protein family. In this study, the authors define molecular determinants of TPD54 association with membranes (a positively charged stretch of amino acids) and homo- and hetero-dimerization (a predicted coiled-coil (CC) sequence). In addition, the authors identify Rab proteins that associate with INVs containing TPD52 and TPD53, as a complement to their previously published identification of Rabs that associate with TPD54-containing INVs. Finally, the authors show that depletion of TPD52-family proteins causes changes in cell shape and migration, and reduces integrin α 5 recycling. The experimental evidence and data analyses presented in the paper are of excellent quality. However, the paper reads like a collection of additional data on TPD52-family proteins that are only tenuously connected to the main conclusion of the paper: that INVs mediate integrin trafficking during cell migration. Moreover, this latter conclusion is insufficiently supported by the data. For example, a detailed study of the role of TPD52-family proteins and INVs in integrin trafficking would require analyses of the recycling of other integrins, integrin expression at the cell surface, presence of integrins in INVs, etc. Additional concerns are described below.

The Reviewer is correct that our previous title implied that we had established that INVs mediated recycling of all known integrin heterodimers which is not the case. Our study investigated the role INVs in recycling of α 5 β 1 integrins mainly and so we have changed the title to *Intracellular nanovesicles mediate α 5 β 1 integrin trafficking during cell migration*.

Major comments

1. Fig. 1. The detection of INVs in this study is largely based on indirect methods. It requires for the readers to go back to the authors' previous paper for evidence of TPD protein association with INVs. For example, the association of TPD proteins and their variants with INVs is measured by "spatiotemporal variance of fluorescence", but this method is not explained in the current manuscript. In the videos it is not even clear if the dots are individual nanovesicles or nanodomains of larger organelles. The reader might wonder why the INVs and their associated proteins were not analyzed by STORM, as in the previous study, or biochemically characterized after isolation by subcellular fractionation, as can be done for similarly sized synaptic vesicles.

The Reviewer is correct that our methods for studying INVs are largely indirect, although we do show the INVs directly by electron microscopy in Figures 3 and 5 in the revised paper. Not explaining the spatiotemporal variance and relying on the previous paper was an oversight. We now explain the method in detail and have included a schematic diagram in Figure 1C to show the principle behind it. The first sentence of Results has been changed to say *We previously found that TPD54 is tightly associated with INVs and that its association with these fast-moving sub-resolution vesicles could be measured by spatiotemporal variance of fluorescence microscopy images. To give a better explanation of what the method is about.*

Our conclusions on whether or not a mutant is on INVs is now corroborated by the rerouting and mitochondrial aggregation assay. The short answer to why we do not use STORM for those experiments is that it is much easier to screen mutants using other methods. Moreover, our previous STORM analysis was on CRISPR knock-in cell lines to look at TPD54 on INVs in the native state. We have not generated the necessary lines to examine other proteins in this way. Overexpression would raise a question as to whether the presence of proteins in INVs was artefactual.

The suggestion to purify the INVs is an excellent one and something we are actively pursuing. In the

paper we show a crude IP-based preparation from RPE-1 cells and in addition proximity biotinylation from BioID2 comparison between TPD54 and R159E. Both analyses show evidence for integrins in INVs. We are currently working on a definitive INV proteome. To do this we are pursuing a variety of methods including synaptic vesicle-style purification, using cell lines other than RPE-1. We hope to publish this analysis after extensive validation work as a separate study in the future.

2. Fig. 1. It would be interesting to determine if the 155-206 sequence in TPD54 is sufficient for recruitment to INVs. In addition, it would be worth mentioning that the 83-206 sequence lacking the CC domain is recruited to membranes, in line with the subsequent finding that multimerization mediated by this domain is dispensable for binding to INVs.

Our experiments to address whether 155-206 are sufficient for recruitment to INVs are now part of a large revamp of this whole part of the paper. In brief, we studied four additional truncation constructs (126-180, 126-206, 155-180, 155-206) to test this idea and found that none of them were recruited to INVs (measured by spatiotemporal variance or by INV capture after rerouting to mitochondria). Note that 83-206 is found on INVs. Together with new data from revised liposome-binding experiments (described below), a picture emerges where there are two regions that together allow association with INVs. A “membrane-binding” region between 83-126 and an INV recognition motif in the conserved C-terminal region. Both regions are necessary but neither are sufficient.

We thank the Reviewer for this suggestion which led to us being able to clarify the molecular details of TPD54-INV association.

Thanks to the Reviewer’s other suggestion, we have also added This result is in agreement with the finding that a TPD54 construct lacking the coiled-coil region (83-206) is localized to INVs (Figure 1C and 3D). to the relevant section.

3. The data in Fig. 2D require analysis of statistical significance of differences. Overall, there doesn’t seem to be a strong preference of GST-TPD54 for small liposomes or a strong requirement of R159 for association with liposomes of any size. It is hard to understand how these small differences could explain preferential association of TPD proteins with INVs vs. larger organelles, where these proteins are also present.

This was very useful criticism. The Reviewer is correct that R159E still bound significantly to membrane and this was difficult to explain. In response to Reviewer 2 (point 1), we have significantly expanded the liposome-binding experiments, including redoing the experiments with R159E. In short, we found that membrane binding in this assay is conferred by a region between 83-155 (probably 83-125) and that mutation of the positive charges has no effect on liposome binding. This indicates that this assay does not fully recapitulate INV binding and that we are missing a protein or lipid factor in the *in vitro* assay that is normally present on INVs.

Our new liposome-binding experiments and reanalysis of previous data still suggests a “curvature preference” that is present for wild-type and mutants. We do not place much emphasis on this in the revised paper. The fact that we can observe binding to smaller liposomes is important to show, and we have not removed this data, but we are not confident about any *preference* for high curvature liposomes. Although this assay is standard in the field, it crucially depends on the sedimentation of liposomes which is much less efficient for smaller sized liposomes. We are uneasy about drawing strong conclusions where the assay has the least resolution, especially when we know that it does not faithfully recapitulate INV binding.

4. Figs. 6 and 7. Since TPD proteins are overexpressed in some cancers, it would be important to analyze if TPD protein overexpression causes changes in cell shape and migration, and in integrin trafficking; the effects would be expected to be opposite to those caused by TPD protein knockdown. This would complement the bioinformatic analysis (Fig. S3) and provide support to the hypothesis that up-regulation of TPD proteins enhances the metastatic potential of certain cancers.

In Figure 8D, we now show that overexpression of TPD54 leads to an increase in migration speed of RPE-1 cells on fibronectin. Importantly, overexpression of the R159E mutant – which does not associate with INVs – does not cause an increase and is similar to expression of GFP alone. We also show that this “opposite of knockdown” trend holds true for TPD52 and TPD53 where overexpression also boosts migration (Figure 8G). We thank the Reviewer for suggesting these experiments and agree that they support the hypothesis that upregulation of TPD52-like proteins enhances metastatic potential.

We did not observe any change in cell shape with overexpression of TPD52-like proteins, nor would we expect to. The defect in cell shape that we saw with depletion of TPD54 was caused by abnormal contact with the substrate. Normal migrating control cells have an optimal shape for migration and the cells traveling faster after overexpression have the same optimal shape.

The question about overexpression and integrin trafficking was difficult to answer experimentally since the ELISA-based assay is a population measurement and our transfection efficiency in RPE-1 cells is very low. For migration experiments where the cells are marked by GFP-expression this is less of an issue. Having demonstrated a boost in migration with overexpression, we focussed our efforts on the other requests the Reviewers made.

5. In the cell migration results shown in Figs. 6B, F and H, is there a way to determine the statistical significance of differences? The curves look similar to me, and I don't see how the authors could draw conclusions about differences in directionality. Also, in Figure 6D, I wonder if the authors tried to rescue the TPD54 KD phenotype by overexpressing TPD52 or TPD53, to determine if there is functional redundancy among the members of this family. This could help interpret the low amplification profiles of some of the TPD52-like proteins in the cancer cells analyzed from the PanCancer Atlas.

The Reviewer is quite right that the curves are similar. There are some small differences in the siTPD54 compared to control, but these differences do not carry into siTPD52 and siTPD53. A non-parametric Kolmogorov-Smirnov test, which is probably the best way to test this, gives $p > 0.05$. Accordingly, we have removed these plots from the main paper (they were duplicated in the supplementary information anyway and remain there for the curious reader).

As described above, we have now overexpressed TPD52 or TPD53 on a control background and observe a boost in migration. We tried (extensively) to do the overexpression on a background of TPD54 depletion and failed for technical reasons. RPE-1 cells did not survive the TPD54 depletion and overexpression procedure, no matter what we tried. However, the Reviewer's question about functional redundancy is answered in the revised manuscript, which now contains more data to support this point, including INV capture by each TPD52-like protein in the absence of TPD54, similar depletion phenotypes in migration and invasion, similar overexpression phenotype in 2D migration, overlapping Rabs in INV capture using each TPD52-like protein.

6. The authors should explain the rationale for investigating the trafficking of integrin alpha-5 and not other integrins. The title of the paper seems to imply a general role of INVs in integrin trafficking. In addition, the authors state "TPD54-depleted cells had a strong reduction in migration speed on both substrates, indicating that this defect is not restricted to a single integrin heterodimer." The trafficking of other integrins should be studied to draw such general conclusions.

During the revisions, our case for INV-based trafficking of other integrins has been strengthened, but not to a level that the original title implied. In brief, we have added new proteomic data that shows a number of integrin types in INVs. First, an IP of TPD54-positive membrane structures that shows TPD52-like proteins, Rabs and integrins. Second, a BioID analysis of WT vs R159E (cannot localize to INVs) which identified a smaller set of Rabs and integrins. We also added imaging of $\alpha 5$ integrin uptake and recycling in control and TPD54-depleted cells to complement the ELISA-based assay. Although several integrin types were identified in the proteomic analyses and a migration phenotype was also observed on laminin (implicating $\alpha 3$ or $\alpha 6$), our manuscript only assesses recycling of $\alpha 5\beta 1$. Therefore, we have changed the

title to *Intracellular nanovesicles mediate $\alpha 5\beta 1$ integrin trafficking during cell migration*.

7. An in-depth analysis of the role of TPD proteins in integrin trafficking requires measurement of integrin levels at the cell surface and a demonstration that integrins localize to INVs. Could it be that the role of TPD proteins in integrin trafficking is mediated by association with organelles other than INVs?

Our response above partially answers this comment. In addition, related to the amount of $\alpha 5$ integrin on the surface, we have now included more details in the paper: The initial surface label and uptake was similar in control and TPD54-depleted cells (98% and 84% of control, respectively). Aberrant traffic of internalized $\alpha 5$ integrin uptake in TPD54-depleted RPE-1 cells could be seen by labeling surface integrins and monitoring uptake and subsequent recycling (Figure 10D). TPD54-depleted cells showed accumulation of labeled integrins in intracellular compartments after 90 min consistent with a defect in recycling. Again, the initial surface (−30 min) and uptake (0 min) pools were similar (Figure 10D).

It is true that TPD54 is localized to other organelles besides INVs and they might be involved in integrin trafficking. However, the sheer number of TPD54-positive INVs is at least an order of magnitude greater so it seems most likely that the vesicles that are their main localization should be invoked to explain the phenotypes observed.

8. The first sentence of the "Discussion" section should be changed to "...we demonstrated the involvement of TPD52-like proteins in integrin trafficking...", since I don't see direct evidence that INVs are the responsible for that role.

In light of our revisions and to be consistent with the title of the paper, we have changed this sentence to: ...we demonstrated the involvement of INVs in $\alpha 5\beta 1$ integrin trafficking and cell migration.

Minor comments:

9. Fig. 3B. It would be helpful to show the input and immunoprecipitated amounts of the FLAG-TPD proteins co-expressed with the GFP-TPD54 proteins.

We have added the input and immunoprecipitated amounts of FLAG-tagged TPD52-like proteins to the Figure (now Figure 4B).

10. In the "Introduction" section, there is a typo: "In breast cancer, TPD52 overexpression correlates with poor prognosis a decrease in metastasis-free survival". It should read "In breast cancer, TPD52 overexpression correlates with poor prognosis and decrease in metastasis-free survival".

Thanks, corrected.

11. Authors should mention in the "Methods" section, or at least in the legend to Fig. 1/S1, how they predicted the presence of coiled coils in these proteins.

We used PCOILS and this is now stated in the legends for Figure 1 and Figure 4.

12. Page 2. In the sentences "Mutation of either K154, R159, K165 and K175/K177 to glutamic acid reduced the variance to levels similar to control (Figure 1B). Whereas similar mutations (K15E or R95E) outside this region had no significant effect" the period should be changed to a comma.

This sentence has been changed in the new version.

13. Page 4, there are two typos in the same sentence that starts with "As with TPD54, Rab30 was the strongest hit with TPD52 (Figure 4B) and TPD53 4D,...". Please, correct with "As with TPD54, Rab30 was the strongest hit with TPD52 (Figure 4B) and TPD53 (Figure 4D),...".

Thanks, corrected.

14. Please, define XFP in the legend to Figs. 2E and S2A schemes, or replace it by mCherry (Fig. 2E) and GFP (Fig. S2A).

Added. Figure S2A has been incorporated into the main figures (Figures 3 and 5). Figure 3 has the schematic diagram and still shows a mix of GFP and mCherry, so we have defined XFP in the legend.

15. Legend to Fig. 4. Define F_{post}/F_{pre} or direct the reader to the "Methods" section to find its definition.

We have added to the legend (now Figure 5): F_{post}/F_{pre}, the average fluorescence of indicated GFP-Rab at mitochondria after rerouting (post) divided by the fluorescence in the same region(s) before (pre).

16. Page 6. In the sentence "We found that TPD54-depleted cells had a strong reduction in migration speed on both substrates" say that depletion was by siRNA treatment.

We have changed this part of the sentence to read "We found that cells depleted of TPD54 by RNAi had a strong reduction...".

17. Page 6, section "Amplification of TPD52-like proteins...". The penultimate sentence needs to end with a parenthesis ")".

Thanks, added.

18. In the sentence, "We compared cells transfected with siCtrl expressing GFP with those transfected with siTPD54 that re-expressed GFP or GFP-TPD54 WT (Figure 6 D)" explain that the re-expressed construct was siRNA-resistant.

Thanks, added.

19. In the "Discussion" section, there's a missing "s" after "heterodimer" in the question: "...how many integrin heterodimer can be trafficked in an INV?".

Thanks, corrected.

20. Also, in the last sentence of the "Discussion" section, when claiming that the "maximum capacity of an INV is surprisingly high", authors may consider mentioning the actual numbers.

We have not added numbers since the capacity depends on multiple factors, as explained in the cited paper. In light of this comment, we have changed this part of the sentence to read *maximum theoretical capacity* of an INV is surprisingly high to make it clear that it is the theoretical capacity and not an experimentally determined capacity that is studied in the cited paper.

21. Similar to TPD54, indicate the ID (GeneBank, Uniprot) of the isoform used in the cloning of TPD52 and TPD53 constructs.

Added - to Methods and to the legend for Figure 1.

22. Indicate the backbone/vector used to obtain the GST-tagged constructs.

Added - pGEX-6P-1.

June 28, 2021

RE: JCB Manuscript #202009028R

Prof. Stephen J Royle
University of Warwick
Centre for Mechanochemical Cell Biology Division of Biomedical Cell Biology Warwick Medical School
Gibbet Hill Road
Coventry CV4 7AL
United Kingdom

Dear Stephen,

Thank you for revising your manuscript entitled "Intracellular nanovesicles mediate $\alpha 5\beta 1$ integrin trafficking during cell migration" based on the reviewer feedback from Journal of Cell Biology. We attempted to send the manuscript back to all three original reviewers. However, unfortunately only reviewer#1 was available to re-review.

As you can see, reviewer#1 was satisfied with your revisions. We have now carefully assessed your responses to the other two reviewers. We appreciate your efforts in addressing the concerns carefully and with new experimentation, especially considering the challenges posed by the pandemic and ensuing restrictions. It seems that you have satisfactorily also addressed the concerns of reviewers #2 and #3 and we would be happy to publish your paper in JCB pending final revisions necessary to meet our formatting guidelines (see details below). We hope that you will agree that the rigorous JCB reviewing has resulted in a superior manuscript about which you can be justifiably proud.

A. MANUSCRIPT ORGANIZATION AND FORMATTING:

Full guidelines are available on our Instructions for Authors page, <https://jcb.rupress.org/submission-guidelines#revised>. **Submission of a paper that does not conform to JCB guidelines will delay the acceptance of your manuscript.**

1) Text limits: Character count for Articles is < 40,000, not including spaces. Count includes title page, abstract, introduction, results, discussion, acknowledgments, and figure legends. Count does not include materials and methods, references, tables, or supplemental legends.

2) Figures limits: Articles may have up to 10 main text figures.

3) Figure formatting: Scale bars must be present on all microscopy images, * including inset magnifications (you may alternatively indicate the diameter of the insert in the figure legend). Molecular weight or nucleic acid size markers must be included on all gel electrophoresis.

4) Statistical analysis: Error bars on graphic representations of numerical data must be clearly described in the figure legend. The number of independent data points (n) represented in a graph must be indicated in the legend. Statistical methods should be explained in full in the materials and methods. For figures presenting pooled data the statistical measure should be defined in the figure legends. Please also be sure to indicate the statistical tests used in each of your experiments (either in the figure legend itself or in a separate methods section) as well as the parameters of the test (for example, if you ran a t-test, please indicate if it was one- or two-sided, etc.). Also, if you used parametric tests, please indicate if the data distribution was tested for normality (and if so, how). If not, you must state something to the effect that "Data distribution was assumed to be normal but this was not formally tested."

5) Abstract and title: The abstract should be no longer than 160 words and should communicate the significance of the paper for a general audience. The title should be less than 100 characters including spaces. Make the title concise but accessible to a general readership.

6) Materials and methods: Should be comprehensive and not simply reference a previous publication for details on how an experiment was performed. Please provide full descriptions in the text for readers who may not have access to referenced manuscripts.

7) Please be sure to provide the sequences for all of your primers/oligos and RNAi constructs in the materials and methods. You must also indicate in the methods the source, species, and catalog numbers (where appropriate) for all of your antibodies. Please also indicate the acquisition and quantification methods for immunoblotting/western blots.

8) Microscope image acquisition: The following information must be provided about the acquisition and processing of images:

a. Make and model of microscope

b. Type, magnification, and numerical aperture of the objective lenses c. Temperature

d. Imaging medium

e. Fluorochromes

f. Camera make and model

g. Acquisition software

h. Any software used for image processing subsequent to data acquisition. Please include details and types of operations involved (e.g., type of deconvolution, 3D reconstitutions, surface or volume rendering, gamma adjustments, etc.).

10) Supplemental materials: There are strict limits on the allowable amount of supplemental data. Articles may have up to 5 supplemental display items (figures and tables). Please also note that tables, like figures, should be provided as individual, editable files. A summary of all supplemental material should appear at the end of the Materials and methods section.

12) Conflict of interest statement: JCB requires inclusion of a statement in the acknowledgements

regarding competing financial interests. If no competing financial interests exist, please include the following statement: "The authors declare no competing financial interests." If competing interests are declared, please follow your statement of these competing interests with the following statement: "The authors declare no further competing financial interests."

13) ORCID IDs: ORCID IDs are unique identifiers allowing researchers to create a record of their various scholarly contributions in a single place. At resubmission of your final files, please consider providing an ORCID ID for as many contributing authors as possible.

B. FINAL FILES:

Thank you for this interesting contribution, we look forward to publishing your paper in Journal of Cell Biology.

Sincerely,

Johanna Ivaska
Monitoring Editor

Andrea L. Marat
Senior Scientific Editor

Journal of Cell Biology

Reviewer #1 (Comments to the Authors (Required)):

The authors have responded in a satisfactory manner to the comments that I had on the initial version of the manuscript.